# PRIVACY-PRESERVING TASK-AGNOSTIC VISION TRANSFORMER FOR IMAGE PROCESSING

## ABSTRACT

Distributed collaborative learning approaches such as federated and split learning have attracted significant attention lately due to their ability to train neural networks using data from multiple sources without sharing data. However, they are not usually suitable in applications where each client carries out different tasks with its own data. Recently, Vision Transformer (ViT) has been widely explored in computer vision applications due to its capability to learn the common representation through global attention of the embedded input sequence. By leveraging the advantages of ViT, here we present a new distributed learning framework for image processing tasks, allowing clients to learn multiple tasks with their private data. The key idea arises from a disentangled representation of local and non-local features using a task-agnostic Vision Transformer and a task-specific head/tail. By connecting task-specific heads and tails at client sides to a task-agnostic Transformer body at a server side, each client learns a translation from its own task to a common representation, while the Transformer body learns global attention between the features embedded in the representation. To enable decomposition between the task-specific and common representation, we propose an alternating training strategy in which task-specific learning for the heads and tails is run on the clients by fixing the Transformer, which alternates with task-agnostic learning for the Transformer on the server by freezing the heads and tails. Once the Transformer body is fully trained with a sufficient number of tasks and clients, additional training of the Transformer body is no longer required when a new client is added with a new task, and all that is required is the training of customer-specific head and tail. Experimental results on multi-task learning for various low-level and high-level computer vision including medical image data show that our method synergistically improves the performance of the task-specific network of each client while maintaining privacy.

## 1 INTRODUCTION

Deep learning approaches have demonstrated the state-of-the-art performance and fast inference time in computer vision tasks (Ronneberger et al., 2015; Zhang et al., 2017a; Wang et al., 2017). In particular, convolutional neural networks (CNN) can learn the hierarchy of complex image features, so that a variety of CNN-based methods have been developed for denoising (Zhang et al., 2017b; Chang et al., 2020), deraining (Wei et al., 2019; Ren et al., 2019), deblurring (Nah et al., 2017; Kupyn et al., 2019), deblocking (Li et al., 2020b; Maleki et al., 2018), etc. However, the performance of CNN typically depends on a large number of training data (Chervenak et al., 2000; Krizhevsky et al., 2017), and it is often difficult to collect data from various entities due to privacy and regulation issues (Price & Cohen, 2019). Since the amount of data from a single source may not be enough, a deep learning framework that can leverage many datasets without violating privacy is required in real-world applications.

To address this, distributed collaborative learning (DCL) approaches, which jointly train a single network on multiple systems or devices without revealing distributed data to a central entity or to each device, have been investigated (Konečný et al., 2016; McMahan et al., 2017a; Gupta & Raskar, 2018). For example, federated learning (FL) (McMahan et al., 2017a; Li et al., 2020c) is studied to aggregate all data to the center under privacy constraints. Thanks to the parallel communication between each client, FL enables fast training of the network across multiple clients. Also, split learning (SL) (Gupta & Raskar, 2018; Vepakomma et al., 2018) is developed as an enhanced privacy-preserving model that

splits a network into clients and server so that each client does not share all network parameters but only train a part of networks. From the advantages of FL and SL, a combination of split and federated learning, named SplitFed learning (SFL) (Thapa et al., 2020), has been recently proposed to provide efficient training and a high level of privacy with a less computational burden. However, the existing CNN-based methods are difficult to determine the proper layer of the network to split. Also, although training data are distributed across each client, all clients usually consider a common learning task.

Meanwhile, in many practical image processing applications, it is unlikely that all the clients are interested in the same applications. For example, some of the clients may be interested in image denoising (Zhang et al., 2017b), whereas the other clients are focused on image deblurring (Nah et al., 2017), deraining (Wei et al., 2019), deblocking (Li et al., 2020b), etc. As each task is different from the others, the existing distributed learning framework may not work. That said, these image processing tasks still require understanding of common image representation, so one may wonder whether there is any systematic way of synergistically learning multiple image processing tasks in a privacy-preserving manner.

One of the most important contributions of this work is to show that *Task-agnostic Vision Transformer* (TAViT), composed of the CNN-based head and tail and ViT-based body, is nicely fit to this purpose. Specifically, the head and tail are placed on each client to learn specific image processing tasks, while the body is stored and trained on a server to learn common representation across all tasks of clients. In contrast to the existing SL framework where the network split is arbitrary, TAViT provides a systematic way of splitting neural networks between clients and servers for privacy-preserving training without losing any performance. Furthermore, TAViT allows clients to use a common Transformer body model to learn multiple image processing tasks and synergistically improve the performance of their task-specific networks.

One may think that the proposed method is similar to the image processing transformer (IPT) (Chen et al., 2020), which consists of CNN-based heads and tails and a Transformer body. However, IPT requires centralized data and large computation resources for both pretraining and task-specific fine-tuning the whole model. Also, the Transformer in IPT has an encoder-decoder architecture which needs an explicit conditioning vector to convert the Transformer for a specific task. Thus, to our best knowledge, IPT is not suitable for distributed learning. In contrast, the body of TAViT is made of an encoder-only Transformer architecture to learn global embedding features of multiple tasks without any condition. Besides, by imposing computation of this Transformer body on the server rather than clients, our framework enables clients to reduce the computational burden while maintaining the overall performance for specific image processing tasks.

In addition, TAViT views the heads and tail at the clients and the body at the server as two-part players and updates them alternately. Specifically, our training step is composed of task-specific learning and task-agnostic learning: the former is to train the client-side heads and tails to learn each task of the client, while the latter is to train the server-side Transformer body to learn general feature interpretation over multiple tasks. When there are more than two clients for any single task, parameters of their heads and tails can be aggregated through FL. Accordingly, TAViT offers seamless integration between SL and FL approaches to protect privacy.

Recall that one of the most unique advantages of Transformer body is to convert "unattended " input features into "attended " output features by learning global attention and non-local interactions between the input features. Accordingly, with the help of aforementioned alternating training scheme, the task-specific head/tail can be only trained to learn task-specific local features, whereas the global feature can be learned through the Transformer. In fact, this disentangled representation of local and non-local features has been pursued throughout the development of deep networks (Ye et al., 2018; Zhang et al., 2019b; Wang et al., 2018). Thus, the proposed Transformer-based approach is considered to be one of the most advanced architectures for achieving this goal, as it improves synergistically overall performance, and at the same time leads the privacy-preserving split learning.

We validate the performance of TAViT on multiple image processing tasks. Experimental results show that our multi-task distributed learning framework using the alternating training strategy outperforms the end-to-end learning of each individual task thanks to the decomposition of the task-agnostic Transformer body and task-specific networks. This suggests that our framework is a promising approach for learning multiple tasks with distributed privacy-sensitive data. In sum, our contributions are summarized as follows:

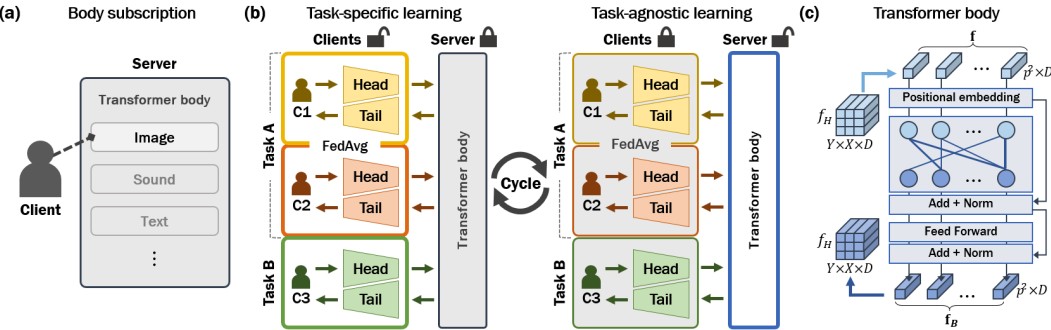

Figure 1: Overall framework of TAViT. (a) By subscribing a task-agnostic Transformer body, (b) clients train their heads and tails with the fixed body in parallel by task-specific learning, while a server trains the body with the fixed head and tail of randomly chosen client for each iteration by task-agnostic learning, (c) and that the body consists of encoder-only Transformer architecture.

- We propose a novel distributed learning framework, TAViT, that carries out multiple image processing tasks using distributed data.
- The proposed method consists of task-specific heads and tails on clients and a task-agnostic Transformer body on a server, which reduces the computational cost of clients and does not require centralized data for multi-task learning.
- An alternating training strategy between the task-specific and task-agnostic learning for the split networks shows the synergy effect of performance improvement, which is demonstrated by experimental results on multiple tasks.

## 2 RELATED WORKS

**Federated learning** In the FL setting, multiple clients learn locally stored data while one server aggregates information of clients by various methods including *FedAvg* (McMahan et al., 2017a). For the efficient implementation of FL, practical challenges of unstable networks, hardware capacity difference, and statistical heterogeneity of data distributions (Li et al., 2020c; Smith et al., 2017; Li et al., 2018) have been actively studied. Corinzia et al. (2019) performs FL with multiple classification tasks, and He et al. (2020) loads a huge network to a server and small CNNs to clients and trains them by knowledge distillation. Yao et al. (2019) presents an unbiased gradient aggregation for FL and meta updating of the model. In contrast, our method is presented for effectively learning on task heterogeneity using distributed data. Although Li et al. (2020a) presents task-agnostic FL method based on the feature extractor, each client trains the task-specific network independently, while our model can learn multiple tasks simultaneously for synergistic performance improvement.

**Split learning** Split learning (SL) is designed to train networks over distributed data by splitting networks into two parts, which updates client-part and server-part networks sequentially (Gupta & Raskar, 2018). By extending this idea, Vepakomma et al. (2018) presents several ways to use SL, and Abuadbba et al. (2020) applies SL to 1D CNN models. However, the existing SL methods are designed using CNN, and to our best knowledge, there is no principle way of splitting the network for the best performance. In particular, Thapa et al. (2020) proposes a combination of FL and SL, but the server requires labels from clients to update the split networks, which may lose data privacy. Also, since outputs are generated from a shared network on the server when there are multiple clients, these methods can be narrowly applied to a single task. In contrast, our model presents Transformer-based shared body that enables multi-task learning of clients without sharing data.

**Vision Transformer for image processing** Recently, inspired by the success of Transformer in natural language processing (Vaswani et al., 2017; Devlin et al., 2018), Transformer-based image processing methods have been extensively explored (He et al., 2021; Han et al., 2021). In particular, Dosovitskiy et al. (2020) proposes a Vision Transformer (ViT) with an encoder-only architecture to learn image recognition tasks. Also, Chen et al. (2020) presents an image processing Transformer (IPT) that learns low-level vision tasks by pretraining and task-specific fine-tuning. However, to the best of our knowledge, there are no existing works that exploit ViT architecture for distributed learning applications.

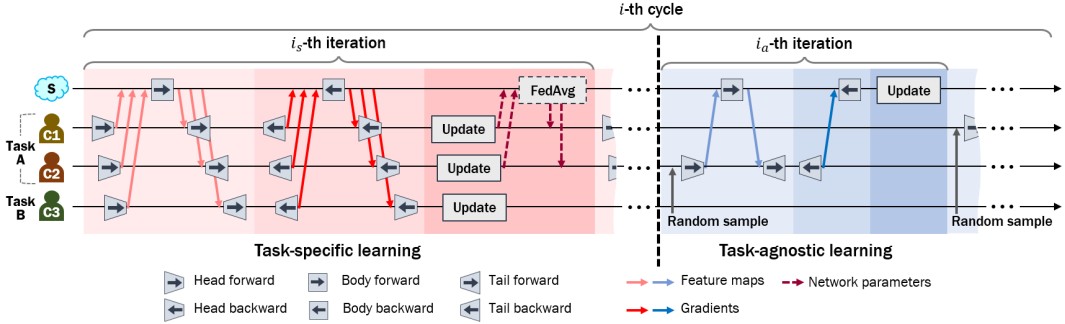

Figure 2: Training scheme of the proposed TAViT, where $S$ denotes a common server, and $C$ denotes clients. The FedAvg with dashed line is applied at the weight aggregation steps.

# 3 PRIVACY-PRESERVING TASK-AGNOSTIC VISION TRANSFORMER

## 3.1 SUBSCRIPTION-BASED SERVICE MODEL

As illustrated in Figure 1(a), TAViT is designed for subscription-based services. Specifically, a client subscribes to a task-agnostic Transformer model at the server side that has learned global attention over the image features from other datasets. Then, the client can build the head and tail proper to its own image processing task, and connect them to the Transformer body at the server. At the subscription time, there may be already multiple clients that subscribe to the same Transformer body. Accordingly, each client can train its own head and tail using its local data whereas the common Transformer body is regularly updated using embedding features from all subscribers through alternating training strategy as shown in Figure 1(b), or even fixed if training has been performed with sufficient number of tasks and clients. As a result, the latest version of the Transformer body trained using more training data can be maintained on the server side so that it can be offered to new clients at the next subscription. Since the local data are not centralized to one device and are not shared with other clients, our framework can preserve data privacy.

In the proposed framework, we consider the features from the head as a sequence of tokens similar to natural language processing. Specifically, as shown in Figure 1(c), we reshape the features $f$ with $Y \times X \times D$ size into a sequence of patches $\mathbf{f} = \{f_1, f_2, \ldots, f_S\}$, where $X, Y, D$ denote width, height, and channel dimension of image features, respectively, $S$ is the number of patches, i.e. $S = YX/p^2$ for patch size $p$, and $f_s$ denotes the $s$-th patch of the features with $p^2 \times D$ size. Then, these reshaped features $\mathbf{f}$ are taken into the Transformer body as an input sequence, which is added to learnable positional embeddings to keep the position information of each feature patch. The Transformer body consists of several encoder layers proposed in Vaswani et al. (2017) so that the encoded features pass through several multi-head self-attention modules and feed-forward modules for each layer. And then, the body output of transformed features is reshaped into the original shape of features $f$ to be used as input for the tail CNN.

Here, for the Transformer body, we employ the encoder-only architecture as a task-agnostic network, compared to IPT (Chen et al., 2020) that uses both encoder and decoder of Transformer. The encoder-only Transformer learns the global relationship between features in the input corpus, and that global attention may be all we need for better performance in vision tasks as demonstrated in ViT. Therefore, the body of our framework can be trained to translate the input embedding features into globally self-attended features independent of specific tasks. Moreover, the heads are guided to learn the task-specific embedding from the input images to the common feature representation, and the tails are trained to learn the attended features for the specific image processing tasks. This architectural modification enables the framework to be suitable for multi-task distributed learning.

## 3.2 TRAINING SCHEME

For distributed datasets of different tasks, we apply the alternating training strategy between clients and the server by considering them as two players. Specifically, as shown in Figure 2, TAViT trains the client-side task-specific head and tail networks and the server-side task-agnostic body network in an alternating manner. In the task-specific learning, clients train their own heads and tails with the fixed body weights in parallel using locally stored datasets. In contrast, in the task-agnostic learning,

**Algorithm 1** TAViT: $\mathcal{C} = \{C_1, C_2, \ldots, C_K\}$ is a group of client sets with different tasks each other. $I_s$ and $I_a$ denote the number of optimization iterations for each task-specific and task-agnostic step in one cycle. $H_c$ and $T_c$ are the head and the tail of a client $c$, and $B$ is the Transformer body on the server.

---

**Initialization :** $H, T$ to all clients, $B$ to a body
**for** $i$ in [1, num_cycles] **do**
   **for** $i_s$ in $[1, I_s]$ **do**                // task-specific learning (heads & tails)
      **for each** client $c \in C_k \subset \mathcal{C}$ **in parallel do**
       |  update $H_c, T_c$ with fixed $B$
      **end**
      **if** $i_s$ is aggregation step **then**     // for case of multi-clients with one task
         **for each** client subset $C_k \subset \mathcal{C}$, s.t. $|C_k| > 1$ **do**
         |  unify $H_c$ and $T_c$ of client $c \in C_k$ (e.g. FedAvg)
         **end**
      **end**
   **end**
   **for** $i_a$ in $[1, I_a]$ **do**                      // task-agnostic learning (body)
      $k \leftarrow$ randomly selected task
      update $B$ with fixed $H_c, T_c$, s.t. $c \in C_k$
   **end**
**end**
**Output:** $H, T, B$

---

the server trains the Transformer body with the fixed head and tail of a randomly chosen client for each iteration. More details are as follows.

### 3.2.1 TASK-SPECIFIC LEARNING

Let $\mathcal{C} = \bigcup_{k=1}^{K} C_k$ be a group of client sets participating different image processing tasks, where $K$ denotes the number of tasks, and $C_k$ has one or more clients with different datasets for the $k$-th task, i.e. $C_k = \{c_1^k, c_2^k, \ldots, c_{N_k}^k\}$ with $N_k \geq 1$. Each client $c \in C_k$ has task-specific own network architecture for a head $H_c$ and a tail $T_c$, which are connected to the Transformer body $B$ in the server.

In the task-specific learning, for the given freezed Transformer $B$ at the server and the local training data $\{(x_c^{(i)}, y_c^{(i)}\}_{i=1}^{N_c}$, the $c$-th client then trains the neural networks $H_c$ and $T_c$ by solving the following optimization problem:

$$\min_{H_c, T_c} \sum_{i=1}^{N_c} \ell_c(y_c^{(i)}, T_c(B(H_c(x_c^{(i)})))), \tag{1}$$

where $\ell_c(y, \hat{y})$ refers to the $c$-th client specific loss between the target $y$ and the estimate $\hat{y}$. The parameters of $H_c$ and $T_c$ are iteratively updated using $\partial \ell_c / \partial T_c$ and $\partial \ell_c / \partial H_c$. These gradients are calculated by back-propagation through the entire model which can be expressed by the chain rule:

$$\frac{\partial \ell_c}{\partial T_c} = \frac{\partial \ell_c}{\partial \hat{y}} \cdot \frac{\partial \hat{y}}{\partial T_c}, \qquad \frac{\partial \ell_c}{\partial H_c} = \frac{\partial \ell_c}{\partial f_H} \cdot \frac{\partial f_H}{\partial H_c} = \frac{\partial \ell_c}{\partial f_B} \cdot \frac{\partial f_B}{\partial f_H} \cdot \frac{\partial f_H}{\partial H_c} \tag{2}$$

where $f_H = H_c(x_c^{(i)})$, $f_B = B(f_H)$ and $\hat{y} = T_c(f_B)$. This implies that to update the head $H_c$ and the tail $T_c$, the gradient $\partial \ell_c / \partial f_B$ is transmitted to the server after back-propagation through the tail, and also the $\partial \ell_c / \partial f_H$ computed from back-propagation through the body is transported to each client.

**Federated learning** In the task-specific learning, when there are multiple clients for the same task $k$ (i.e. $N_k > 1$), their heads and tails can be trained in parallel. Suppose that $c_i^k$ has training dataset size of $|\mathcal{D}_i|$ and the total size of dataset in $C_k$ is $\sum |\mathcal{D}_i| = |\mathcal{D}|$. In this case, the back-propagation and optimization process are the same with the single client case, but additionally applies *FedAvg*(McMahan et al., 2017a) to the parameters $H_c$ and $T_c$ of $c \in C_k$ for every assigned period, which is written as:

$$(H_{c_j}, T_{c_j}) \leftarrow \left( \sum_{i=1}^{N_k} \frac{|\mathcal{D}_i|}{|\mathcal{D}|} H_{c_i}, \sum_{i=1}^{N_k} \frac{|\mathcal{D}_i|}{|\mathcal{D}|} T_{c_i} \right), \quad \text{where } 1 \leq j \leq N_k. \tag{3}$$

The period of the weight aggregation is adjustable (50 epochs in our experiments). From this federated learning, clients corresponding to the $k$-th task share the same parameters at the end of task-specific learning as shown in Figure 2.

### 3.2.2 TASK-AGNOSTIC LEARNING

Once the heads and tails of multiple clients are trained, the Transformer body is trained by fixing the heads and tails at the clients. To train the Transformer body that learns the common representation in a task-agnostic manner, we construct a subset of the client $\mathcal{C}_B$ by selecting one client from each task:

$$\mathcal{C}_B = \{c_{n_1}^1, c_{n_2}^2, \dots, c_{n_K}^K\}, \quad c_{n_k}^k \in C_k. \tag{4}$$

Then, the training data $\{x_c^{(i)}, y_c^{(i)}\}_{i=1}^{N_c}$ corresponding to the task of the client are also selected, and the Transformer body in the server is updated by solving the optimization problem as follows:

$$\min_B \sum_{c \in \mathcal{C}_B} \sum_{i=1}^{N_c} \ell_c(y_c^{(i)}, T_c(B(H_c(x_c^{(i)})))). \tag{5}$$

Similar to the task-specific learning, the parameters of $B$ are updated using $\partial \ell_c / \partial B$, where the client $c$ is randomly chosen from $\mathcal{C}_B$ at each optimization step. The required gradients also come from back-propagation as following:

$$\frac{\partial \ell_c}{\partial B} = \frac{\partial \ell_c}{\partial f_B} \cdot \frac{\partial f_B}{\partial B}, \quad \text{where} \quad \frac{\partial \ell_c}{\partial f_B} = \frac{\partial \ell_c}{\partial \hat{y}} \cdot \frac{\partial \hat{y}}{\partial f_B}, \tag{6}$$

where $f_B = B(f_H)$ and $\hat{y} = T_c(f_B)$. Here, the gradient $\partial \ell_c / \partial f_B$ is only transported to the server after back-propagation through the tail. Through this task-agnostic learning, the Transformer body in the server learns global embedding representation and provides task-agnostic self-attended features for various image processing. The pseudocode of the overall TAViT is described in Algorithm 1 with more details in Appendix A.

### 3.3 COMMUNICATION COST AND PRIVACY PRESERVATION BY TAViT

Given that gradients have to be transmitted two-way or one-way for training head/tail and body parts of the architecture, one many wonder whether additional communication overhead is significant. However, since the Transformer body is a shared model on the server that does not perform any weight aggregation, our model has much smaller cost for one communication between the client and the server in the task-agnostic learning. This comes from the small size of transported features and gradients for the heads and tails. If we sample clients in the network training, the communication cost can be further controlled. Therefore, up to a certain epoch size, our model is more communication bandwidth efficient compared to the classical FL, and the advantage increases if a bigger Transformer body is used for better representation of global attention. For detailed analysis, see Appendix D.4.

The proposed TAViT is designed to use distributed local data for distinct tasks without sharing the data to the other clients or any central device. Although the privacy attack on the transported features between the server and clients can be occurred, yet another powerful and unique mechanism for maintaining privacy in TAViT arises when the client-side network of the proposed method has a skip connection between the head and the tail. In this case, the transported feature characteristics can contain very lossy information from the original data, and one cannot reconstruct data only using the transmitted hidden features of the proposed method as detailed in Appendix D.1.

## 4 EXPERIMENTAL RESULTS

We examine the performance of TAViT with the following image processing tasks: deblocking (JPEG artifact removal), denoising, deraining, and deblurring. Additional experiments for image inpainting and medical data are also performed to investigate its performance for high-level computer vision tasks and different domain data, respectively, which can be found in Section D.5 of Appendix. With a single server, we set two clients to carry out FL on the deblocking task and set one client for each of the other tasks, so the total number of clients is five in our experiments. We evaluate results using two metrics of PSNR and SSIM.

**Datasets** The public datasets we used are as follows. For deblocking and denoising, we use 10,582 images from PASCAL VOC 2012 (Everingham et al., 2010) and Segmentation Boundaries Dataset

(SBD) (Hariharan et al., 2011). Particularly, for FL on the deblocking, we split the data into two sets with 5,291 images and distribute them to each client. Deblocking results are evaluated on Berkely Segmentation Database (BSD500) (Martin et al., 2001b) that provides 200 test images. For the denoising, we apply random Gaussian noise with the level of $\sigma = 30$ to images. The Denoising model is evaluated on CBSD68 that contains 68 test images extracted from the BSD500. For deraining, following the experiment setting of Jiang et al. (2020), we use data from Rain14000 (Fu et al., 2017b), Rain1800 (Yang et al., 2017), Rain800 (Zhang et al., 2019a), and Rain12 (Li et al., 2016) that provide 13,711 pairs of clean and synthetic rain images. Deraining results are evaluated on Rain100H and Rain100L (Yang et al., 2017), each of which has 100 synthetic rainy images. Deblurring is performed using a GoPro dataset (Nah et al., 2017) that contains 2,103 and 1,111 pairs of sharp and blurry images for training and test sets, respectively.

**Implementation details**  To implement our TAViT, we use the encoder and decoder of DDPM (Ho et al., 2020) with three stages as our backbone of each head and tail at the client. For the Transformer body in the server, we use 8 encoder layers of the vanilla Transformer (Vaswani et al., 2017) with 256 words and 512 embedding dimensions. The total number of parameters for networks at each client and the server is about 28M and 17M, respectively. Using 4 Nvidia Quadro RTX 6000 cards and 2 Nvidia Geforce GTX 1080Ti cards, we train the networks using Adam optimizer with a learning rate of $3 \times 10^{-5}$. We initialize parameters of the networks with those of the pre-trained model by an autoencoder scheme. For the data augmentation, we apply random horizontal and vertical flipping, rotating with 90 degrees, and cropping by a patch size of $64 \times 64 \times 3$. For three cycles, by setting the batch size as 20, we perform the task-specific learning for 200, 400, 400, and 2000 epochs on deblocking, denoising, deraining, and deblurring, respectively. Also, we perform the task-agnostic learning for 1000 epochs with $1/4$ data for each task. We implement our TAViT using PyTorch library under BSD-style license using *Flower* federated learning protocol (Beutel et al., 2020) under Apache 2.0 License. The details of datasets and implementation are described in Appendix B.

## 4.1  RESULTS

**Convergence of TAViT for multi-task distributed learning**  We evaluated the results of the proposed TAViT of multi-task distributed learning with all participated clients and one common Transformer body in the server. Figure 3 shows the gradual progression of quality metrics through the alternating training scheme. The performance of all tasks from our method increased and outperformed as the task-specific learning and the task-agnostic learning continued. This demonstrates the synergistic improvement of the task-specific heads/tails and the task-agnostic body: the heads and tails learn more accurate feature embedding for given tasks, and the common body learns the global attention general to multiple image processing tasks by looking at various datasets. Although there were some tasks in which the score of each step was slightly lower than that of the previous step by the interaction of different task datasets, the overall performance of TAViT was improved as the cycle progressed. Detailed quantitative results for each cycle are described in Appendix C.

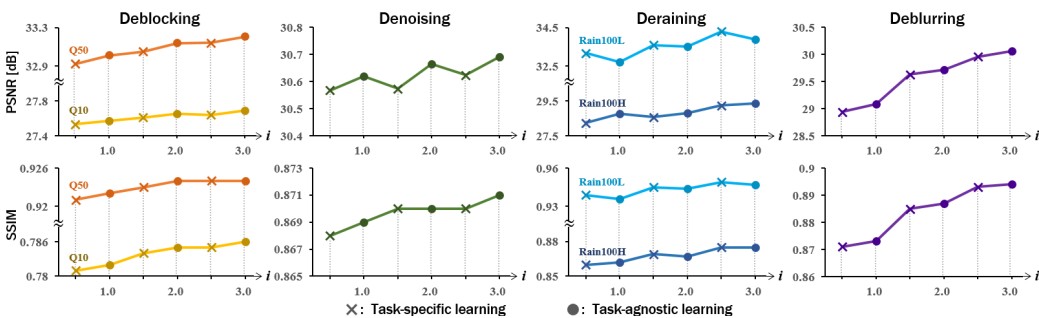

Figure 3: Results of TAViT for multi-task distributed learning. Each column shows the PSNR and SSIM according to $i$-th cycle for the deblocking, denoising, deraining, and deblurring, respectively.

**Comparison of TAViT to other strategies**  We compared our TAViT with other distributed learning strategies: SL and FL. We conducted both SL and FL with the two clients assigned for the deblocking task. For SL, as designed in Vepakomma et al. (2018), we placed the head and tail networks on clients and the body on the server, and trained those split networks without the weight aggregation for the head and tail. For FL, we placed the entire model composed of the head, body, and tail on each

client, and trained the network in parallel by carrying out the aggregation with *FedAvg* (McMahan et al., 2017a) using a common server. Figure 4 shows these scenarios, where C1 and C2 are clients for the deblocking, C3, C4, and C5 are clients for the denoising, deraining, deblurring, and S is the server. As reported in Table 1, the proposed method achieves better performance compared to the other strategies even though ours learns multiple tasks.

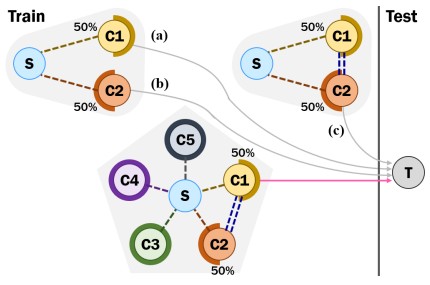

Figure 4: Scenarios of TAViT (pink arrow) and other strategies of (a-b) SL and (c) FL, where C# is clients and S is the server.

Table 1: Comparison result of our TAViT to other distributed learning strategies. Q# denotes the quality of JPEG images for deblocking task.

| Method | Device | Q10 | | Q50 | |
|---|---|---|---|---|---|
| | | PSNR | SSIM | PSNR | SSIM |
| Split learning (SL) | C1, S | 27.55 | 0.781 | 32.97 | 0.921 |
| Split learning (SL) | C2, S | 27.51 | 0.781 | 32.88 | 0.921 |
| Federated learning (FL) | C1, C2, S | 27.46 | 0.780 | 32.79 | 0.919 |
| TAViT (Ours) | C1, C2, S | **27.69** | **0.786** | **33.20** | **0.924** |

**Comparison of TAViT to learning each separate task**    To verify the capability of the task-agnostic Transformer body learning from multiple tasks, we compared TAViT with the models independently trained on each individual task. Under the setting of centralized data for each task, we implemented this study in two manners: end-to-end learning (EL) and single-task learning (STL). For EL, we trained the whole network in one device through the end-to-end optimization scheme. For STL, we distributed the decomposable head, body, and tail to a client and a server as the proposed method, and trained the networks by the alternating training strategy for one cycle. Table 2 reports the results on Benchmark datasets for each task. This shows that our TAViT trained on multiple tasks simultaneously outperforms both EL and STL, which suggests that the task-agnostic body of our framework does not degrade the model by task heterogeneity but enhances the performance for various tasks.

**Comparison of TAViT to CNN-based models**    To compare the performance of TAViT with CNN-based deep learning models, we tested several existing methods on benchmark datasets for each task. Table 2 and Figure 5 show the qualitative and visual comparison results, respectively. For the deblocking, when comparing with DnCNN Zhang et al. (2017a), AR-CNN Dong et al. (2015), and QCN (Li et al., 2020b), the proposed method outperforms for both the 10 and 50 levels of quantization quality. Visual comparisons also show that the proposed method removes block artifacts clearly compared to the others. For the denoising, we compared our method with CBM3D (Dabov et al., 2007), DnCNN (Zhang et al., 2017a), FFDNet (Zhang et al., 2018b), IRCNN (Zhang et al., 2017b), DHDN (Park et al., 2019), and SADNet (Chang et al., 2020). The results show that our

Table 2: Comparison results on Benchmark datasets. For Transformer-based methods, EL is the end-to-end learning, STL is the single-task learning, and ours is the multi-task learning using TAViT.

| Task | | Method | | | | | | | | | |
|---|---|---|---|---|---|---|---|---|---|---|---|
| Dataset | Metric | | | CNN-based | | | | | | Transformer-based | |
| **Deblocking** | | Input | DnCNN | | AR-CNN | | QCN | | EL | STL | Ours |
| BSDS500 | PSNR | 25.67 | 26.70 | | 26.42 | | 27.66 | | 27.67 | 27.59 | **27.69** |
| (Q10) | SSIM | 0.719 | 0.755 | | 0.777 | | **0.811** | | 0.785 | 0.785 | 0.786 |
| BSDS500 | PSNR | 31.51 | 32.70 | | - | | 33.00 | | 33.01 | 32.93 | **33.20** |
| (Q50) | SSIM | 0.902 | 0.918 | | - | | **0.934** | | 0.923 | 0.924 | 0.924 |
| **Denoising** | | Input | CBM3D | DnCNN | FFDNet | IRCNN | DHDN | SADNet | EL | STL | Ours |
| CBSD68 | PSNR | 19.03 | 29.71 | 30.32 | 30.31 | 30.22 | 30.41 | 30.64 | 30.43 | 30.65 | **30.69** |
| (σ=30) | SSIM | 0.336 | 0.843 | 0.861 | 0.860 | 0.861 | 0.864 | - | 0.864 | 0.870 | **0.871** |
| **Deraining** | | Input | DerainNet | SEMI | UMRL | PreNet | MSPFN | | EL | STL | Ours |
| Rain100H | PSNR | 12.13 | 14.92 | 16.56 | 26.01 | 26.77 | 28.66 | | 28.88 | 28.95 | **29.35** |
| | SSIM | 0.349 | 0.592 | 0.486 | 0.832 | 0.858 | 0.860 | | 0.863 | 0.864 | **0.875** |
| Rain100L | PSNR | 25.52 | 27.03 | 25.03 | 29.18 | 32.44 | 32.40 | | 32.93 | 32.50 | **34.30** |
| | SSIM | 0.825 | 0.884 | 0.842 | 0.923 | **0.950** | 0.933 | | 0.937 | 0.935 | 0.949 |
| **Deblurring** | | Input | DeblurGAN | Nah et al. (2017) | | Zhang et al. (2018a) | | DeblurGANv2 | EL | STL | Ours |
| GoPro | PSNR | 25.64 | 28.70 | 29.08 | | 29.19 | | 29.55 | 28.62 | 29.28 | **30.06** |
| | SSIM | 0.790 | 0.858 | 0.914 | | 0.931 | | **0.934** | 0.864 | 0.877 | 0.894 |

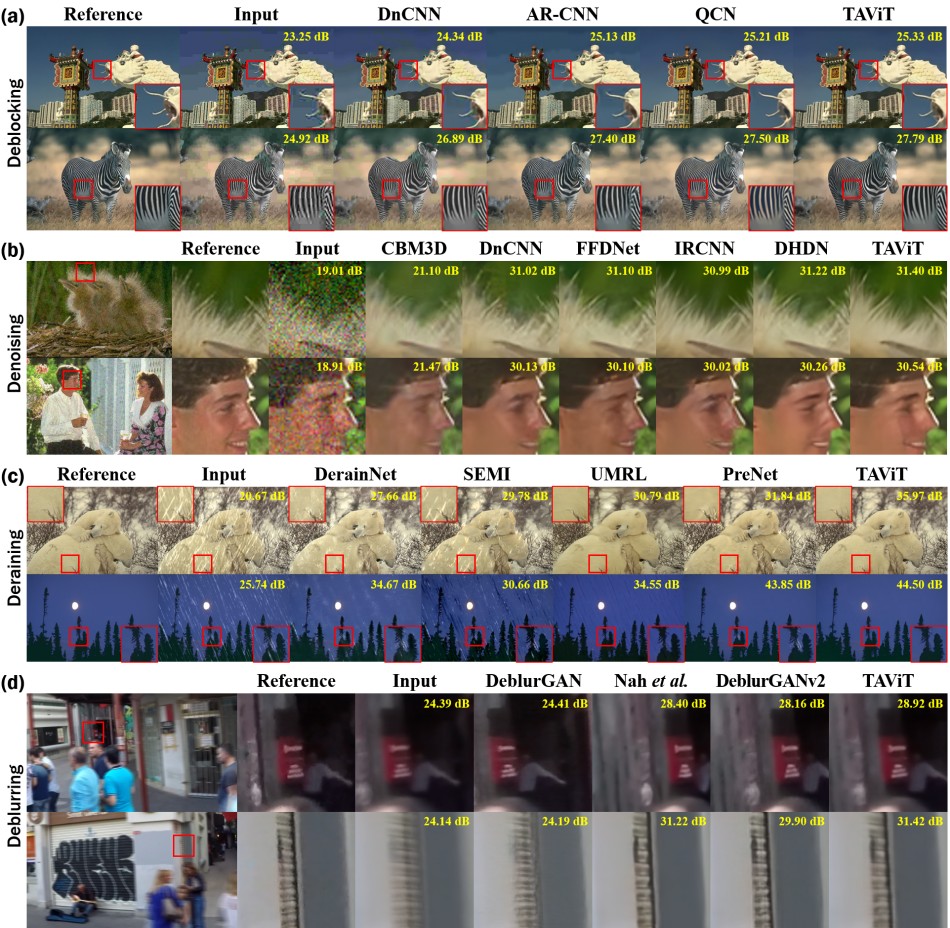

Figure 5: Visual comparisons for multiple tasks: (a) deblocking, (b) denoising, (c) deraining, and (d) deblurring. Yellow values are PSNR, and inset box is a magnified view of a red rectangle.

TAViT achieves better PSNR/SSIM scores, and also provides more clearly denoised images while preserving structure and texture details than the comparisons. For the deraining, we tested our model in addition to DerainNet (Fu et al., 2017a), SEMI (Wei et al., 2019), UMRL (Yasarla & Patel, 2019), PreNet (Ren et al., 2019), and MSPFN (Jiang et al., 2020). We used Y channel in YCbCr color space followed by Jiang et al. (2020) for the evaluation. As a result, our model outperforms the comparative methods on both Rain100H and Rain100L. Also, the images restored by ours are closer to the references by removing rain streaks than the others. For the deblurring, we employed DeblurGAN (Kupyn et al., 2018), Nah et al. (2017), Zhang et al. (2018a), DeblurGANv2 (Kupyn et al., 2019) for comparisons. The results show that the proposed method achieves comparable performance to the existing approaches. Visual results show that our TAViT restores blurry images with sharp edges, while the others still contain blurry artifacts or position shifting of objects compared to the references.

## 5 CONCLUSION

In this work, we present a multi-task distributed learning framework called TAViT. In TAViT, the task-specific head CNN and the tail CNN are distributed to clients that have their own data, which are connected to a common Transformer body placed in the server. With an alternating training scheme, the heads and tails on client sides are trained by task-specific learning, while the body is trained by task-agnostic learning. We conduct experiments on four different image processing tasks, which shows the success of task-agnostic learning of the Transformer body and its synergistic improvement with the task-specific heads and tails. Through our model, the participating clients can design and train their own networks depending on the task using local data in parallel. We expect that the proposed TAViT can be efficiently used in the case that sharing data with other institutions is sensitive such as medical fields.

**Ethics statement**    As our work utilizes distributed learning models, similar to the existing FL and SL, our method may be vulnerable to privacy attacks against the server such as inversion attacks (Yin et al., 2021). Although the proposed framework is designed by encoding the feature maps and gradients under Flower protocol which makes it difficult for attackers to restore the original data, the hidden feature may leak the raw data to some degree. Thus, privacy-related techniques such as differential privacy (McMahan et al., 2017b) and authenticated encryption of data (Rogaway, 2002) should be analyzed for practical applications.

**Reproducibility statement**    The source code and our trained models to reproduce the proposed method are available at `https://github.com/TAViT2022/TAViT`. For the detailed pseudocode, refer to Appendix A. Also, the data processing steps for datasets used in the experiments are provided in Appendix B.

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

# A    DETAILS OF TAViT WITH PSEUDOCODE

As described in the main paper, the task-specific heads and tails in clients and the Transformer body in the server are trained in an alternating manner between the proposed task-specific learning and the task-agnostic learning. In the following, we describe each step in more detail in terms of its implementation.

**Pseudocode for the task-specific learning**    Algorithm 2 shows the pseudocode for the task-specific learning. Given $K$ image processing tasks, the task-specific learning updates the heads $H$ and the tails $T$ in each client with the fixed body $B$. The server first initializes global weights of the heads

---

**Algorithm 2** Task-specific learning of TAViT: $I_s$ denotes the number of iterations for the task-specific learning in one cycle. $(H_c, T_c)$ denote the head and the tail in a client $c \in C_k$ for the $k$-th task, and $B$ denotes the Transformer body in a common server. $(H_{C_k}, T_{C_k})$ are global weights of the heads and the tails for the task $k$. $(f_H, f_B)$ are output feature maps from the head and the body, and $\hat{y}$ is the output of the tail. $\ell_c$ is the task-specific loss of the client $c$. $|\mathcal{D}_j|$ is the size of training data at $c_j$, and $|\mathcal{D}|$ is the total size of training data at $C_k$, i.e. $|\mathcal{D}| = \sum_j |\mathcal{D}_j|$.

---

```
/* run on the server (with fixed B) */
```
**Initialize** $H_{C_k}, T_{C_k}$
**Send** $H_{C_k}, T_{C_k}$ to all clients $c \in C_k$
**for** $i_s$ in $[1, I_s]$ **do**
    **for each** client $c \in C_k$, where $k = \{1, 2, \ldots, K\}$ **in parallel do**
        $f_H \leftarrow$ `ClientPhase1()`
        $f_B \leftarrow B(f_H)$                                       // body output
        $\frac{\partial \ell_c}{\partial f_B} \leftarrow$ `ClientPhase2(`$f_B$`)`
        $\frac{\partial \ell_c}{\partial f_H} \leftarrow \frac{\partial \ell_c}{\partial f_B} \cdot \frac{\partial f_B}{\partial f_H}$            // backpropagation through body
        `ClientUpdate(`$\frac{\partial \ell_c}{\partial f_H}$`)`
    **end**
    **if** $i_s$ is weight aggregation step for $|C_k| > 1$ **then**
        **Get** $(H_{c_j}, T_{c_j})$ from client $c_j$, where $j \in \{1, 2, \ldots, N_k\}$
        $H_{C_k} \leftarrow \sum_{j=1}^{N_k} \frac{|\mathcal{D}_j|}{|\mathcal{D}|} H_{c_j}$            // FedAvg for head
        $T_{C_k} \leftarrow \sum_{j=1}^{N_k} \frac{|\mathcal{D}_j|}{|\mathcal{D}|} T_{c_j}$             // FedAvg for tail
        **Send** $(H_{C_k}, T_{C_K})$ to all clients $c \in C_k$
    **end**
**end**

```
/* run on client c */
```
**Function** `ClientPhase1()`
    $x, y \leftarrow$ set current data and label
    $f_H \leftarrow H_c(x)$                                           // head output
    **return** $f_H$

```
/* run on client c */
```
**Function** `ClientPhase2(`$f_B$`)`
    $\hat{y} \leftarrow T_c(f_B)$                                      // tail output
    $\ell_c \leftarrow Loss(y, \hat{y})$
    $\frac{\partial \ell_c}{\partial T_c} \leftarrow \frac{\partial \ell_c}{\partial \hat{y}} \cdot \frac{\partial \hat{y}}{\partial T_c}$        // computation of tail gradients
    $\frac{\partial \ell_c}{\partial f_B} \leftarrow \frac{\partial \ell_c}{\partial T_c} \cdot \frac{\partial T_c}{\partial f_B}$
    **return** $\frac{\partial \ell_c}{\partial f_B}$

```
/* run on client c */
```
**Function** `ClientUpdate(`$\frac{\partial \ell_c}{\partial f_H}$`)`
    $\frac{\partial \ell_c}{\partial H_c} \leftarrow \frac{\partial \ell_c}{\partial f_H} \cdot \frac{\partial f_H}{\partial H_c}$        // computation of head gradients
    update $H_c, T_c$ using $\frac{\partial \ell_c}{\partial H_c}$ and $\frac{\partial \ell_c}{\partial T_c}$ by optimizer e.g. Adam

---

---

**Algorithm 3** Task-agnostic learning of TAViT: $I_a$ denotes the number of optimization iterations for task-agnostic learning in one cycle. $(H_c, T_c)$ denote the head and tail in a client $c$ in a client $c \in C_k$ for the $k$-th task, and $B$ denotes the Transformer body in a common server. $(f_H, f_B)$ are output feature maps from the head and the body, and $\hat{y}$ is the output of the tail. $\ell_c$ is the task-specific loss of the client $c$.

---

```
/* run on the server */
```
**Initialize** $\mathcal{C}_B = \{c_{n_1}^1, c_{n_2}^2, \ldots, c_{n_K}^K\}$ where $c_{n_k}^k \in C_k$
**for** $i_a$ in $[1, I_a]$ **do**

    $c \leftarrow c_{n_k}^k \in \mathcal{C}_B$                     `// Random selection of client with task k`

    $f_H \leftarrow$ `ClientPhase1()`

    $f_B \leftarrow B(f_H)$                            `// body output`

    $\frac{\partial \ell_c}{\partial f_B} \leftarrow$ `ClientPhase2`$(f_B)$

    $\frac{\partial \ell_c}{\partial B} \leftarrow \frac{\partial \ell_c}{\partial f_B} \cdot \frac{\partial f_B}{\partial B}$            `// computation of body gradients`

    update $B$ using $\frac{\partial \ell_c}{\partial B}$ by optimizer e.g. Adam

**end**

```
/* run on client c (with fixed Hc, Tc) */
```
**Function** `ClientPhase1()`

    $x, y \leftarrow$ set current data and label

    $f_H \leftarrow H_c(x)$                         `// head output`

    **return** $f_H$

```
/* run on client c (with fixed Hc, Tc) */
```
**Function** `ClientPhase2`$(f_B)$

    $\hat{y} \leftarrow T_c(f_B)$                      `// tail output`

    $\ell_c \leftarrow Loss(y, \hat{y})$

    $\frac{\partial \ell_c}{\partial f_B} \leftarrow \frac{\partial \ell_c}{\partial \hat{y}} \cdot \frac{\partial \hat{y}}{\partial f_B}$        `// backpropagation through tail`

    **return** $\frac{\partial \ell_c}{\partial f_B^c}$

---

and the tails and sends them to all clients in $C_k$, where $C_k$ is a set of clients with different datasets for the $k$-th task. When each client $c \in C_k$ takes local training data $x$ and provides a feature map $f_H$ by the head $H_c$ to the server (line 5 with `ClientPhase1`), the server-side Transformer body takes the feature map $f_H$ as an input embedding and estimates the self-attended features $f_B$ that is independent of specific tasks. Once the $f_B$ in the server is sent to the client, the client computes the task-specific loss $\ell_c$ between the label $y$ and the tail output $\hat{y}$ (line 24 in `ClientPhase2`). The gradient of the tail $\partial \ell_c / \partial T_c$ is also computed in the client at `ClientPhase2` (line 25), which is used to compute $\partial \ell_c / \partial f_B$ that is transported to the server. Then, in the server, $\partial \ell_c / \partial f_H$ is calculated by backpropagation through the body, which is sent to the client so as to compute the head gradient $\partial \ell_c / \partial H_c$ and finally update $H_c$ and $T_c$ (lines 28-30 of `ClientUpdate`) using a single optimizer.

Here, when there are multiple clients for the $k$-th task, i.e. $|C_k| > 1$, we apply the federated learning for the heads and the tails of those clients as described in lines 11-16 in Algorithm 2. The heads and the tails are trained in parallel, and their weights are aggregated by *FedAvg* (McMahan et al., 2017a) on the server side for every weight aggregation period. Then, these updated global weights of the heads $H_{C_k}$ and the tails $T_{C_k}$ and are transmitted to all clients in $C_k$ so that the clients train their own head and tail using the new global weights from the next step.

**Pseudocode for the task-agnostic learning**   In the task-agnostic learning, the Transformer body in the server is updated with the fixed heads and tails of clients. Algorithm 3 shows the pseudocode for the task-agnostic learning of TAViT. Given a subset of clients, $\mathcal{C}_B$, by selecting one client among $C_k$ for each task, the client $c \in \mathcal{C}_B$ is randomly chosen for every iteration. Then, compared to the task-specific learning, the implementation of the task-agnostic learning is similarly conducted but does not need `ClientUpdate` process in Algorithm 2. In other words, after the gradient $\partial \ell_c / \partial f_B$ is computed on the client side at `ClientPhase2` (line 14-18) and transmitted to the server (line 6), the server updates the Transformer body by computing the body gradients $\partial \ell_c / \partial B$ (lines 7-8), which is the final step of each iteration.

# B  DETAILS OF DATASETS AND IMPLEMENTATION

## B.1  LICENSE/SOURCE FOR EACH DATASET

In our experiments, we use the public datasets for image deblocking, denoising, deraining, and deblurring tasks. Here, we describe the specific information of each data set such as license and source link.

**PASCAL VOC 2012**  The PASCAL VOC data set (Everingham et al., 2010) is publicly available, which includes images obtained from the "flickr" website under SmugMug or its third-party licensors. The data are protected by the United States and international intellectual property laws. The data source is from the URL: `http://host.robots.ox.ac.uk/pascal/VOC/`.

**BSDS500 and CBSD68**  The Berkeley Segmentation Data Set and Benchmarks 500 (BSDS500) (Arbeláez et al., 2011) data set is an extended version of BSDS300 (Martin et al., 2001b), which is a public data set originally provided for image segmentation and boundary detection by Berkeley Computer Vision Group. This data set is widely used for measuring image restoration performance. The color BSD68 data set (CBSD68) is extracted from the BSDS500. The BSDS500 can be downloaded at `https://www2.eecs.berkeley.edu/Research/Projects/CS/vision/grouping/resources.html`.

**Synthetic rainy images**  The synthetic rainy data set for training is collected from Rain14000 synthesized by Fu et al. (2017b), Rain1800 authored by Yang et al. (2017), Rain800 created by Zhang et al. (2019a), and Rain12 made by Li et al. (2016). We test our method on the synthetic rainy data sets of Rain100H and Rain100L, both of which are authored by Yang et al. (2017). All these data sets are publicly available and can be downloaded at the following links:

- *Rain14000*: `https://xueyangfu.github.io/projects/cvpr2017.html`
- *Rain1800*: `https://www.icst.pku.edu.cn/struct/Projects/joint_rain_removal.html`
- *Rain800*: `https://github.com/hezhangsprinter/ID-CGAN`
- *Rain12*: `https://yu-li.github.io/`
- *Rain100L & Rain100H*: `https://www.icst.pku.edu.cn/struct/Projects/joint_rain_removal.html`

**GoPro**  The GoPro dataset (Nah et al., 2017) provides training and test sets for deblurring. The data are available at `https://seungjunnah.github.io/Datasets/gopro.html`.

## B.2  DATA PROCESSING

All datasets we used in experiments provide natural images that have three RGB channels and pixel values with a range of [0, 255]. Upon these datasets, we performed the following data processing according to the image processing tasks.

For the image deblocking task, we quantized the images following JPEG compression. We first transformed RGB image into YUV color space using the following equations.

$$Y = 0.257R + 0.504G + 0.098B + 16 \tag{7}$$
$$U = -0.148R - 0.291G + 0.439B + 128 \tag{8}$$
$$V = 0.439R + 0.368G - 0.071B + 128 \tag{9}$$

Then, we split the image into 8x8 blocks without overlapping and apply Discrete Cosine Transform (DCT) to each block. According to the level of quantization quality, we divided each element of the DCT coefficients by proper predefined matrices. After that, we apply inverse DCT and aggregate all blocks into an image, and then, we transformed the image from YUV to RGB color space.

$$R = 1.164(Y - 16) + 1.596(V - 128) \tag{10}$$
$$G = 1.164(Y - 16) - 0.392(U - 128) - 0.813(V - 128) \tag{11}$$
$$B = 1.164(Y - 16) + 2.017(U - 128) \tag{12}$$

For the denoising task, we added Gaussian noise to the clean images. Specifically, we applied random Gaussian noise with the level of $(\mu, \sigma) = (0, 30)$ to images, and then clipped the values into [0, 255].

For the other tasks, the datasets named Rain# and GoPro provide synthetic rainy images and blurry images, respectively. Since we used these datasets for the deraining and deblurring tasks, we did not perform any preprocessing such as the synthesis of rain artifacts blurry effect.

After the above data processing for all tasks, we randomly cropped the images by a patch size of $64 \times 64 \times 3$. Also, we applied data augmentation using random flipping and rotating with 90 degrees. Then, we normalized the images with the scale of pixel values from [0,255] to [-1, 1], which are final inputs to the model.

## B.3 NETWORK ARCHITECTURES

For the task-specific head and tail for each task, we use the network architecture of DDPM (Ho et al., 2020) that is composed of residual blocks and attention modules. We set the number of two-times downsampling/upsampling stages as 3. For each stage, the channel size is set as 128, 256, and 512, respectively. Accordingly, given an input image $x \in \mathbb{R}^{64 \times 64 \times 3}$, the head provides a feature map $f_H \in \mathbb{R}^{16 \times 16 \times 512}$ that passes through the body, and the tail generates an output of the same size as the input.

On the other hand, for the Transformer body, we use the encoder part of the vanilla Transformer (Vaswani et al., 2017). As described in the main paper, the Transformer body takes a sequence of patches $\mathbf{f}$ by reshaping the feature map $f_H$ as an embedding of the words. In the experiments, the length of the input sequence is 256 by setting the patch size as 1, and the sequence dimension is 512. Then, once the input sequence is added to learnable positional encodings, the encoded features $h$ pass through $L$ encoder layers ($L = 8$ in our experiments). Table 3 shows the structure of each encoder layer of the body.

Table 3: The Transformer body architecture and its parameters in our experiments. For each encoder layer $l$, $MHA$ is the multi-head attention modules, $LN$ is the layer-normalization, $DropOut$ is the dropout layer, $Linear$ is the fully-connected layer, and $ReLU$ is the ReLU activation function.

| Encoder layer | Parameters | | |
|---|---|---|---|
| $a^l = MHA(h^l) \in \mathbb{R}^{n \times d}$ | Notation | Value | Meaning |
| $u^l = LN(h^l + DropOut(a^l)) \in \mathbb{R}^{n \times d}$ | $L$ | 8 | The number of encoder layers |
| $v_1^l = ReLU(Linear(u^l)) \in \mathbb{R}^{n \times d_h}$ | $n$ | 256 | The length of sequence |
| $v_2^l = DropOut(Linear(v_1^l)) \in \mathbb{R}^{n \times d}$ | $d$ | 512 | The sequence dimension |
| $h^{l+1} = LN(u^l + v_2^l) \in \mathbb{R}^{n \times d}$ | $d_h$ | 1024 | The hidden dimension |

**Model sizes** Table 4 shows the model sizes of the task-specific head and tail, and the Transformer body we used. When comparing the number of parameters and the size of networks, we can observe that the client-side networks composed of the head and the tail is larger than the task-agnostic Transformer body. Considering the experimental results in the main paper, this model size suggests that the body estimates task-agnostic self-attended features that provide the synergy effect in the task-specific and task-agnostic learning even if the body size is smaller than the sum of head and tail.

Table 4: Model sizes of the head, body, and tail in our experiment.

| Network | Head | Body | Tail |
|---|---|---|---|
| # Parameters | 22,341,891 | 16,822,272 | 5,610,629 |
| Size (MB) | 103.54 | 64.98 | 39.00 |

## C  EXPERIMENTAL RESULTS

### C.1  TAVIT ON MULTIPLE IMAGE PROCESSING TASKS

**Evaluation results of TAViT**  Table 5 reports the quantitative evaluation results of TAViT on multiple image processing tasks, which is visualized with graphs of scores for the cycles in the main paper. Figure 6 shows the qualitative results of TAViT. This shows that the performance of each task is improved according to the cycles between the task-specific and task-agnostic learning.

Table 5: Quantitative results of TAViT according to the cycles, which are visualized with graphs in the main paper. The best results are highlighted in bold.

| Task | Deblocking | | | | Denoising | | Deraining | | | | Deblurring | |
|---|---|---|---|---|---|---|---|---|---|---|---|---|
| Cycle | BSDS500 (Q10) | | BSDS500 (Q50) | | CBSD68 ($\sigma = 30$) | | Rain100H | | Rain100L | | GoPro | |
| | PNSR | SSIM | PNSR | SSIM | PNSR | SSIM | PNSR | SSIM | PNSR | SSIM | PNSR | SSIM |
| 0.5 | 27.53 | 0.781 | 32.92 | 0.921 | 30.57 | 0.868 | 28.24 | 0.860 | 33.17 | 0.939 | 28.94 | 0.871 |
| 1.0 | 27.57 | 0.782 | 33.01 | 0.922 | 30.62 | 0.869 | 28.75 | 0.862 | 32.69 | 0.936 | 29.09 | 0.873 |
| 1.5 | 27.61 | 0.784 | 33.05 | 0.923 | 30.57 | 0.870 | 28.57 | 0.869 | 33.58 | 0.945 | 29.63 | 0.885 |
| 2.0 | 27.65 | 0.785 | 33.14 | 0.924 | 30.66 | 0.870 | 28.79 | 0.867 | 33.50 | 0.944 | 29.72 | 0.887 |
| 2.5 | 27.64 | 0.785 | 33.14 | 0.924 | 30.62 | 0.870 | 29.25 | 0.875 | **34.30** | **0.949** | 29.96 | 0.893 |
| 3.0 | **27.69** | **0.786** | **33.21** | **0.924** | **30.69** | **0.871** | **29.35** | **0.875** | 33.88 | 0.947 | **30.06** | **0.894** |

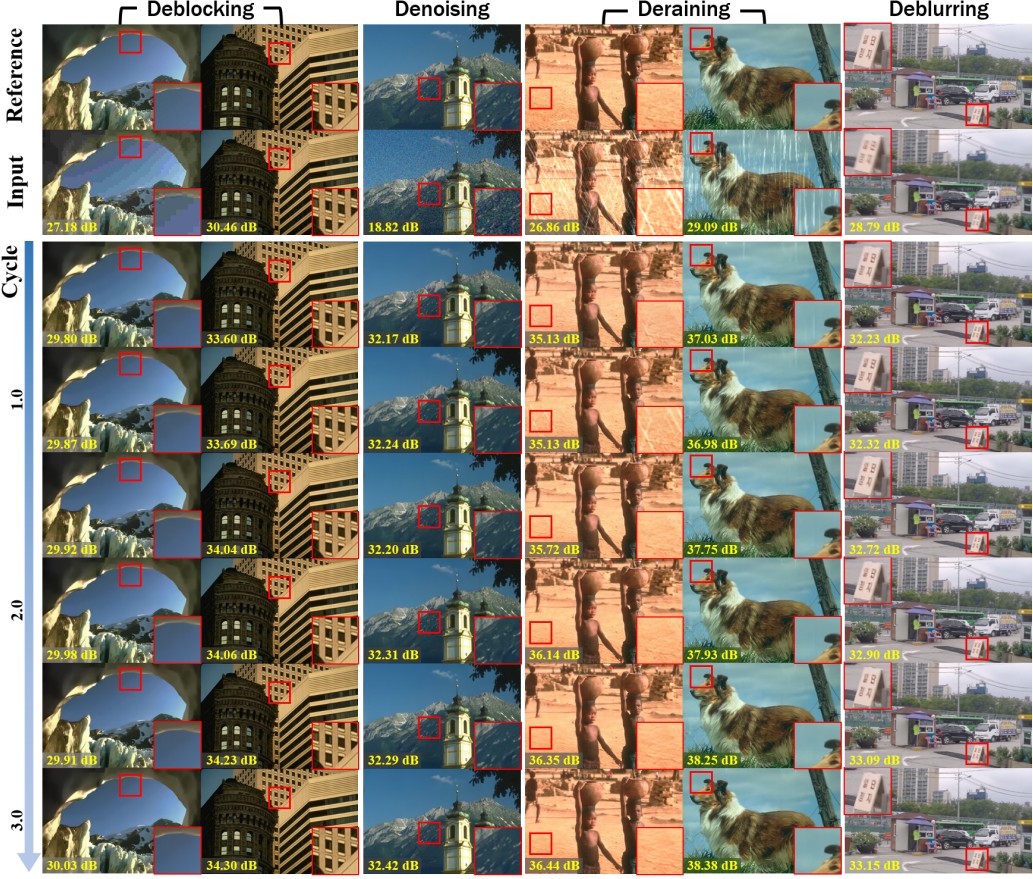

Figure 6: Qualitative results of TAViT according to the cycles. From the left to the right columns, the deblocking results on images with the quantization quality 10 and 50, the denoising results, the deraining results on Rain100H and Rain100L, and the deblurring results. The yellow value is PSNR, and an inset box is a magnified view of a red rectangle.

**Qualitative comparisons** Besides the results presented in the main paper, here, we show more visual comparisons of our TAViT to the existing methods. Figure 7, 8, 9, and 10 display the deblocking, denoising, deraining, and deblurring results, respectively. All these results verify that our TAViT as a distributed learning for multiple image processing tasks outperforms the comparisons.

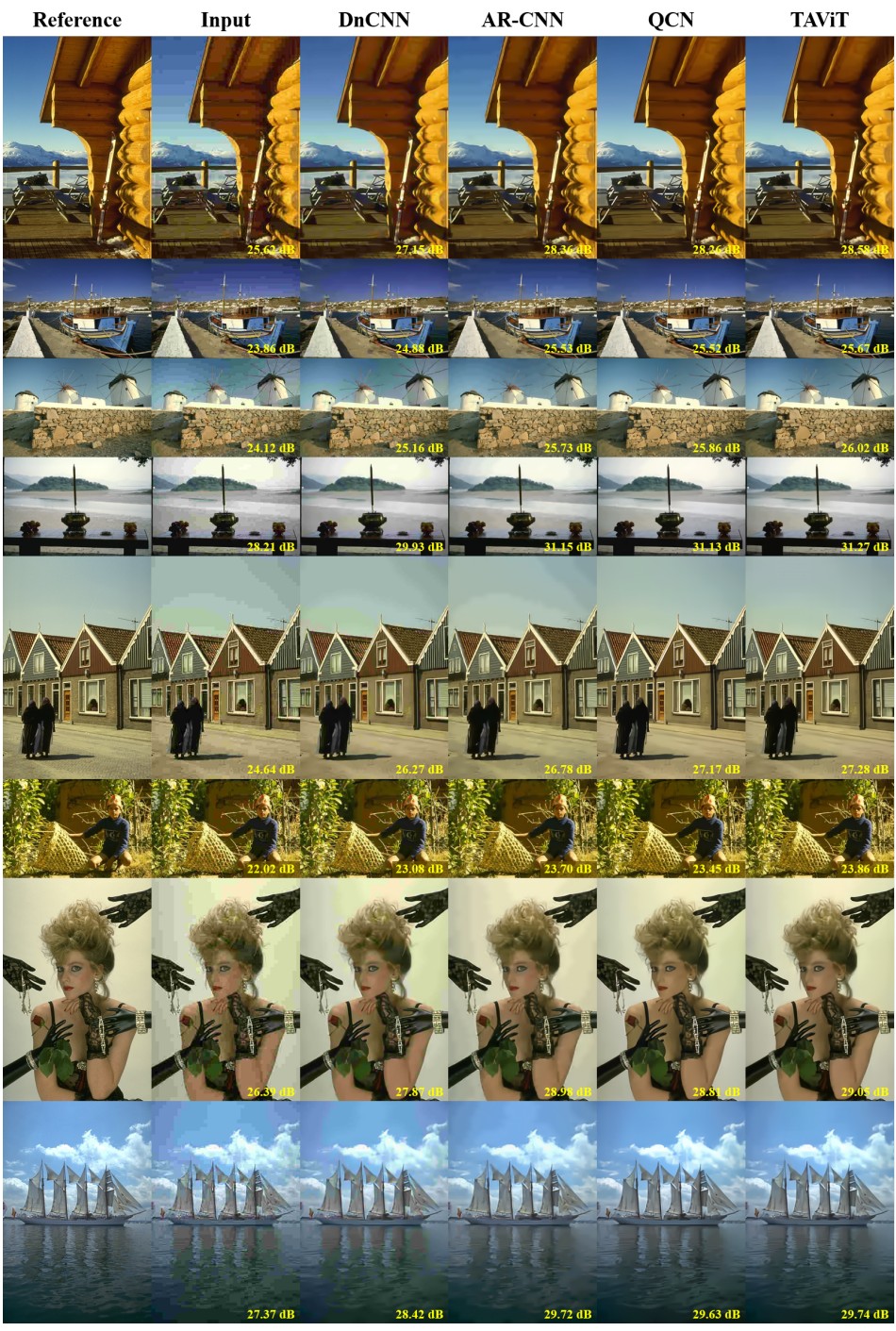

Figure 7: Visual comparisons of image deblocking task using BSDS500 (Arbeláez et al., 2011). The yellow value of each result is PSNR. We compare our TAViT with DnCNN (Zhang et al., 2017a), AR-CNN (Dong et al., 2015), and QCN (Li et al., 2020b).

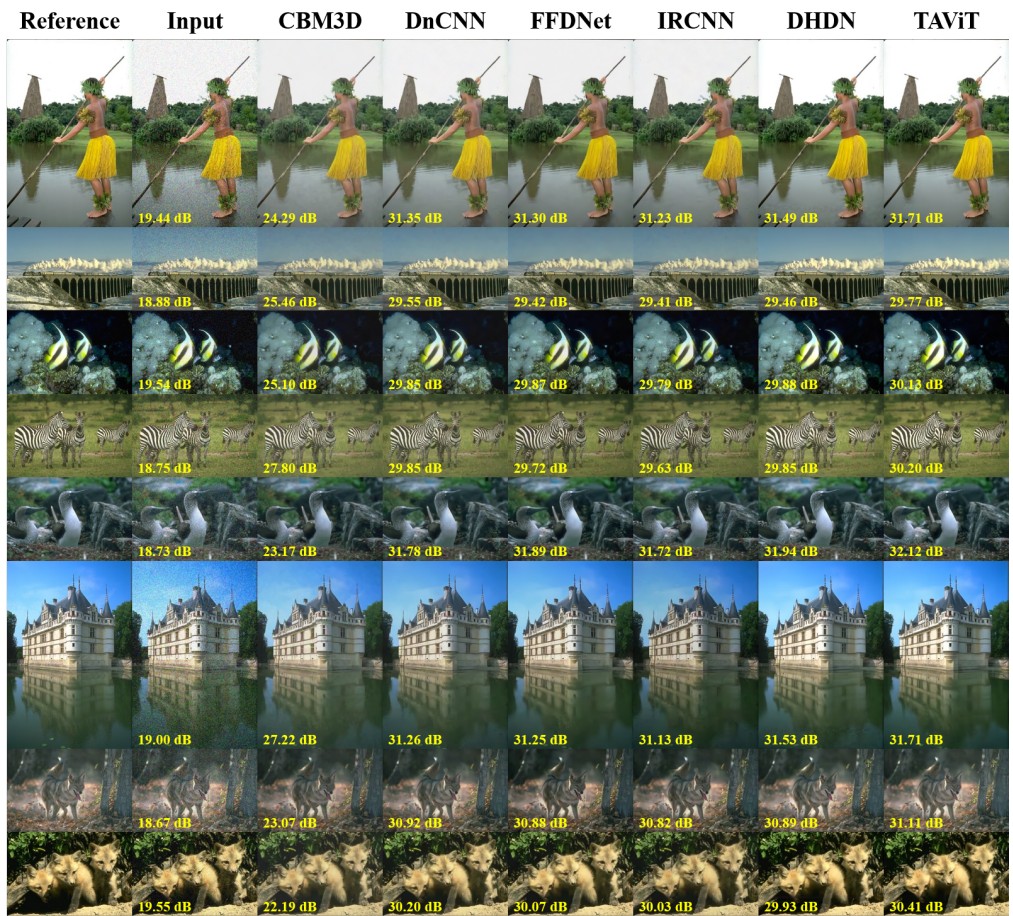

Figure 8: Visual comparisons of image denoising task using CBSD68 (Martin et al., 2001a). The yellow value of each result is PSNR. We compare the proposed TAViT with CBM3D (Dabov et al., 2007), DnCNN (Zhang et al., 2017a), FFDNet (Zhang et al., 2018b), IRCNN (Zhang et al., 2017b), and DHDN (Park et al., 2019).

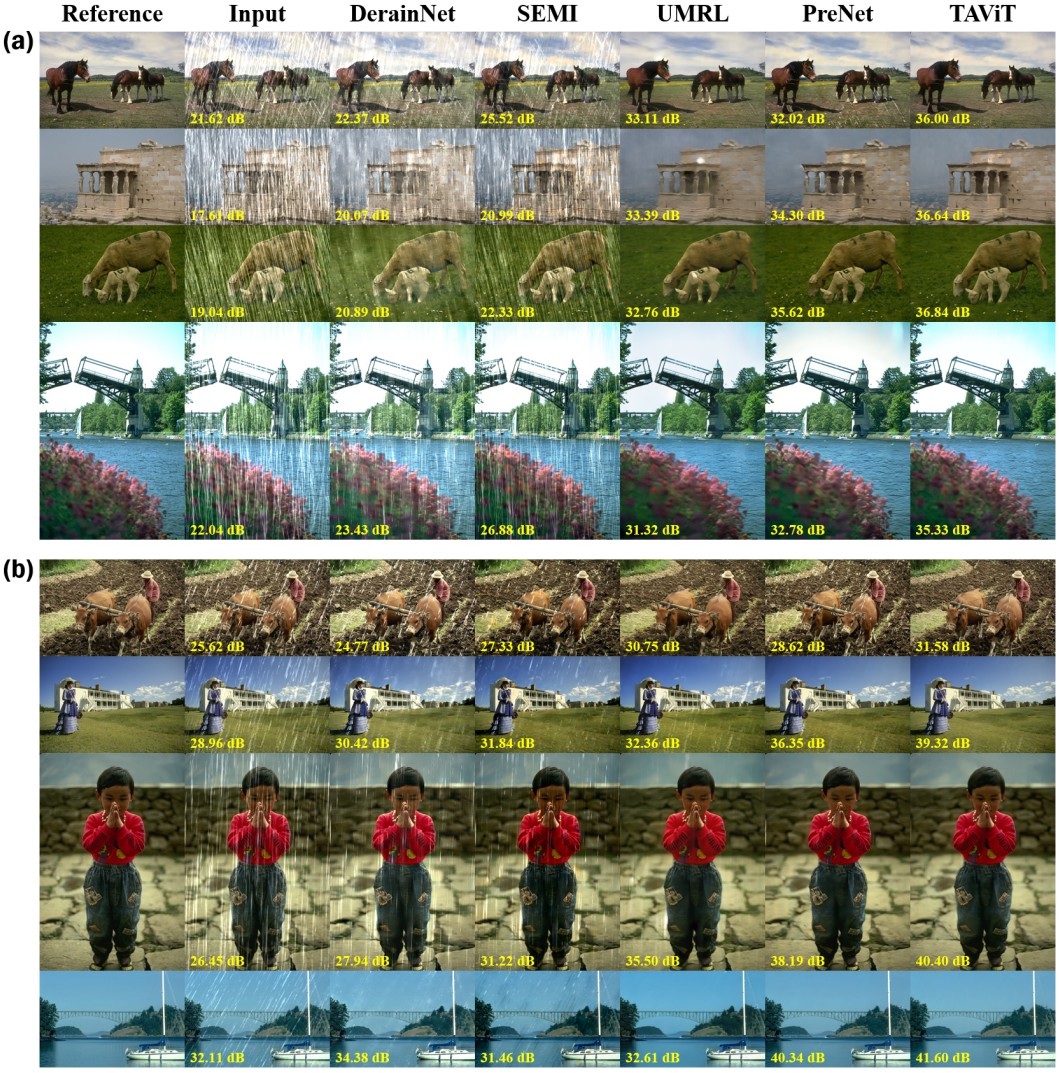

Figure 9: Visual comparisons of image deraining task using Rain100H and Rain100L data sets (Yang et al., 2017). The yellow value of each result is PSNR. DerainNet (Fu et al., 2017a), SEMI (Wei et al., 2019), UMRL (Yasarla & Patel, 2019), and PreNet (Ren et al., 2019) are used to compare our TAViT. (a) Results on Rain100H data set. (b) Results on Rain100L data set.

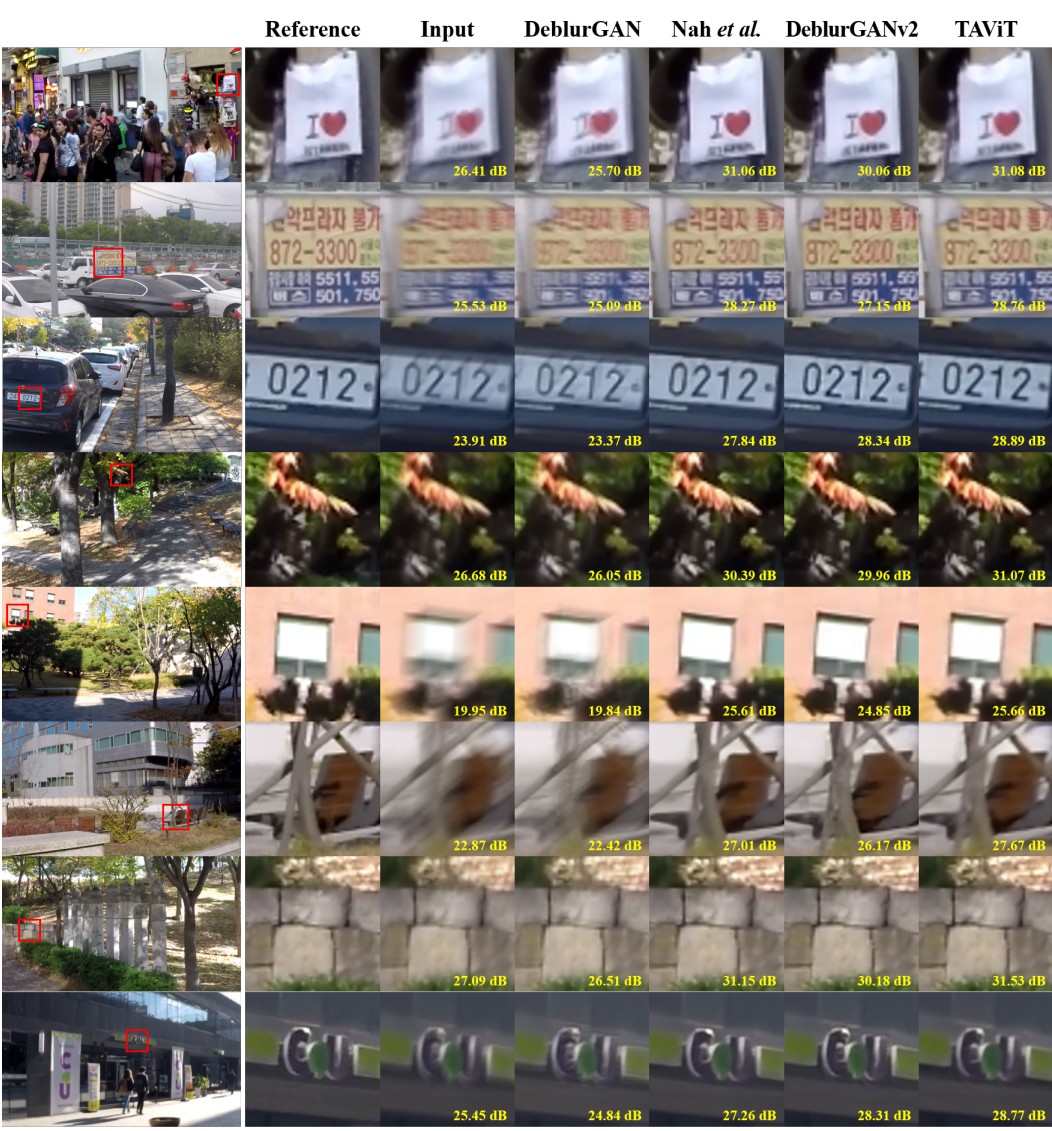

Figure 10: Visual comparisons of image deblurring task using GoPro (Nah et al., 2017) dataset. For visualization, we enlarge a region of red inset box (left column) for all images and denote PSNR of each result with yellow. We compare our TAViT with DeblurGAN (Kupyn et al., 2018), Nah et al. (2017), and DeblurGANv2 (Kupyn et al., 2019).

## C.2 ABLATION STUDY OF TAViT

**Study on the amount of data for each task in the task-agnostic learning**   In the main paper, we implemented our method using 1/4 of the dataset for each task in the task-agnostic learning. To verify that this amount of data is enough for the task-agnostic learning, we performed the ablation study using different amounts of data with a 1/2 ratio for each deblocking, denoising, deraining, and deblurring task. Table 6 and Figure 11 show the quantitative results of TAViT on the multiple tasks using 1/2 data in the task-agnostic learning. Similar to the results with a 1/4 data ratio, the scores of PSNR and SSIM tend to increase as the cycle continues. When comparing the best results from the 1/4 and 1/2 data ratio, we can observe that performance for each task using even 1/4 amount of data is comparable or better than using 1/2 data. This suggests that using 1/4 of data for each task in the task-agnostic learning is sufficient to train the Transformer body and obtain high performance.

Table 6: Quantitative results of TAViT on multiple image processing tasks with 1/2 data for each task in the task-agnostic learning. The best results are highlighted in bold.

| Task | Deblocking | | | | Denoising | | | | Deraining | | | | Deblurring | |
|---|---|---|---|---|---|---|---|---|---|---|---|---|---|---|
| | BSDS500 (Q10) | | BSDS500 (Q50) | | CBSD68 ($\sigma = 30$) | | Rain100H | | Rain100L | | GoPro | | | |
| Cycle | PNSR | SSIM | PNSR | SSIM | PNSR | SSIM | PNSR | SSIM | PNSR | SSIM | PNSR | SSIM | | |
| 0.5 | 27.53 | 0.781 | 32.92 | 0.921 | 30.57 | 0.868 | 28.24 | 0.860 | 33.17 | 0.939 | 28.94 | 0.871 | | |
| 1.0 | 27.57 | 0.782 | 33.02 | 0.922 | 30.62 | 0.869 | 28.78 | 0.862 | 32.47 | 0.933 | 29.18 | 0.875 | | |
| 1.5 | 27.63 | 0.784 | 33.10 | 0.923 | 30.58 | 0.870 | 28.92 | 0.867 | 33.28 | 0.942 | 29.63 | 0.887 | | |
| 2.0 | 27.65 | 0.785 | 33.15 | 0.924 | 30.67 | 0.870 | 29.18 | 0.870 | 33.71 | **0.946** | 29.83 | 0.890 | | |
| 2.5 | 27.66 | 0.785 | 33.15 | 0.923 | 30.56 | 0.870 | 29.35 | 0.875 | **33.83** | 0.945 | 29.94 | 0.893 | | |
| 3.0 | **27.69** | **0.786** | **33.20** | **0.924** | **30.69** | **0.870** | **29.53** | **0.877** | 33.68 | 0.945 | **30.18** | **0.897** | | |
| * 1/4 data for each task | | | | | | | | | | | | | | |
| Best | **27.69** | **0.786** | 33.21 | 0.924 | 30.69 | 0.871 | 29.35 | 0.875 | 34.30 | 0.949 | 30.06 | 0.894 | | |

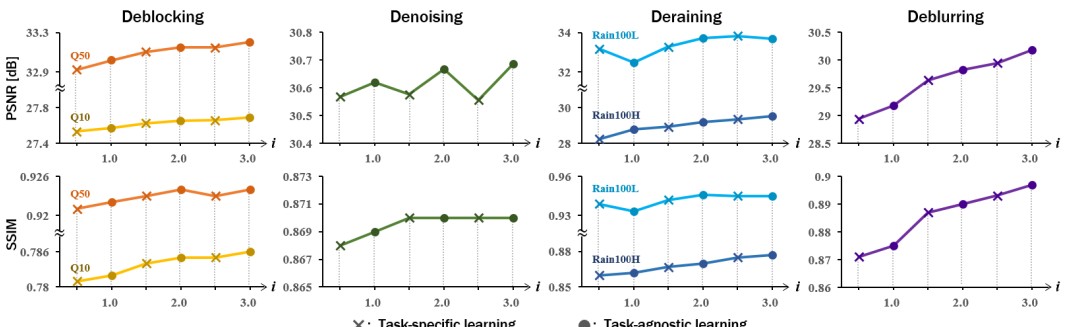

Figure 11: Results of TAViT with 1/2 data for each task in the task-agnostic learning. Each column shows PSNR and SSIM according to $i$-th cycle for the deblocking, denoising, deraining, and deblurring, respectively.

**Study on the weight aggregation period**   In the main paper, we conducted the experiment of TAViT by applying FL to the deblocking task that has two clients with their own data. In FL, the weights of the network in each client are averaged in the server for every weight aggregation period that is given as a hyperparameter. Here, since this period can influence learning performance in that the clients and the common server communicate to aggregate network weights, we performed the ablation study on the weight aggregation period for training the client-side networks. As reported in Table 7, for the deblocking task, we trained the model with the aggregation period of 20, 50, and 100 epochs. When we evaluated the deblocking results, the weight aggregation per 50 epochs provides better performance with 27.53dB/0.781 and 32.92dB/0.921 of PSNR/SSIM for the quality 10 and 50, respectively, compared to the other methods. This verifies that the weight aggregation period of 50 epochs presented in the main paper is proper to train and evaluate the proposed TAViT in our experiments.

Table 7: Results of study on the weight aggregation period of FL for image deblocking task. The best results are highlighted in bold. Q# denotes the quantization quality of JPEG images.

| Aggregation period | Q10 | | Q50 | |
|---|---|---|---|---|
| | PSNR | SSIM | PSNR | SSIM |
| per 20 epochs | 26.53 | 0.769 | 30.91 | 0.910 |
| per 50 epochs | **27.53** | **0.781** | **32.92** | **0.921** |
| per 100 epochs | 26.73 | 0.772 | 31.16 | 0.910 |

# D    DISCUSSION

## D.1    SKIP-CONNECTION OF HEAD AND TAIL FOR PRIVACY PRESERVATION

When configuring the task-specific heads and tails with skip-connections, our model can avoid the privacy attack in some degree while maintaining the encoding information for the tail to generate outputs. This is because the skip-connected features are isolated on each client and not transported to the server. Accordingly, the transported features between the clients and the server may contain far less information about the original data. Figure 12 shows examples of the outputs with and without skip-connections. This shows that the network output without skip-connections barely has the property of original data, which indicates that one may not be able to reconstruct the original data using the transmitted hidden features of the proposed method.

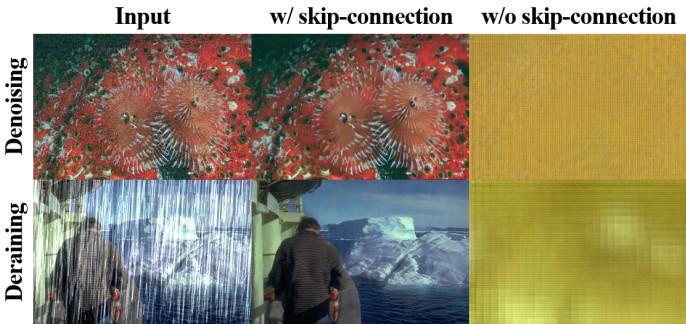

Figure 12: The network output with and without skip-connections between task-specific heads and tail for denoising and deraining task.

## D.2    EFFECT OF TASK-AGNOSTIC TRANSFORMER BODY

As we described in the main paper, the reason for developing our model with CNN-based heads/tails and the Transformer-based body is to take advantage of each network. In particular, Transformer learns the global attention of the input sequence through self-attention modules and has recently been extensively studied for various computer vision tasks. One of the most unique advantages of Transformer is to convert "unattended" input feature vectors into "attended" output feature vectors by learning global attention and non-local interactions between the input features. Accordingly, the task-specific head / tail can be only trained to learn task-specific local features, whereas the global feature can be learned through the Transformer body. This disentangled representation of local and non-local features has been pursued throughout the development of deep networks. Thus, the proposed Transformer-based approach is considered to be one of the most advanced architectures for achieving this goal, as it improves synergistically overall performance, and at the same time leads the privacy-preserving split-learning architecture.

In order to show that this design is proper to the multi-task distributed learning, we additionally conducted the experiment by replacing the Transformer body to the CNN model. Specifically, we configured the CNN body with CBR blocks, where C is a convolutional layer with the consistent channel 512, B is a batch normalization layer, and R is an activation by ReLU layer. For a fair comparison, we set the CBR blocks as 7 to have almost the same number of learnable parameters

with the Transformer body we used (16,522,240 of CNN body vs. 16,953,344 of Transformer body). Then, using this CNN body, we implemented our proposed task-specific and task-agnostic learning for one cycle on the multiple image processing tasks as the main paper. As a result, Table 8 shows that our model with the Transformer body achieves higher performance in both task-specific and task-agnostic learning. This indicates that the Transformer can be used as a general task-agnostic body for multi-task learning.

Table 8: Results of study on the effect of the Transformer body of TAViT versus CNN body.

| Body | Cycle | Deblocking | | | | Denoising | | Deraining | | | | Deblurring | |
| | | BSDS500 (Q10) | | BSDS500 (Q50) | | CBSD68 ($\sigma = 30$) | | Rain100H | | Rain100L | | GoPro | |
| | | PNSR | SSIM | PNSR | SSIM | PNSR | SSIM | PNSR | SSIM | PNSR | SSIM | PNSR | SSIM |
|---|---|---|---|---|---|---|---|---|---|---|---|---|---|
| CNN | 0.5 | 26.42 | 0.752 | 32.92 | 0.921 | 30.22 | 0.866 | 28.23 | 0.851 | 31.58 | 0.930 | 28.78 | 0.865 |
| | 1.0 | 26.49 | 0.755 | 32.95 | 0.921 | 30.50 | 0.866 | 28.26 | 0.852 | 31.77 | 0.930 | 28.94 | 0.868 |
| Ours | 0.5 | 27.53 | 0.781 | 32.92 | 0.921 | 30.57 | 0.868 | 28.24 | 0.860 | 33.17 | 0.939 | 28.94 | 0.871 |
| | 1.0 | 27.57 | 0.782 | 33.01 | 0.922 | 30.62 | 0.869 | 28.75 | 0.862 | 32.69 | 0.936 | 29.09 | 0.873 |

### D.3  SAMPLING STRATEGY OF CLIENTS

When there are multiple clients for one task in task-specific learning, the task-specific networks of clients are aggregated through the sampling strategy of FedAvg. On the other hand, in task-agnostic learning of the proposed TAViT, one client is sampled for each iteration. Since the networks of clients for the same task are aggregated before the task-agnostic learning, we can readily sample one client for each task. Then, choosing one client for the subset of Eq. (4) can be viewed as sampling one task, which naturally reduces the communication cost.

In fact, the performance of TAViT is not affected by the number of sampled clients in the task-agnostic learning, since the task-agnostic body is updated for sufficient iterations. To demonstrate this, we performed the task-agnostic learning for the four tasks in our experiments by varying the sampling strategy. Table 9 shows the results after training our model for one cycle according to the number of sampled clients in the task-agnostic learning. The results show that sampling one client achieves comparable or higher performance for all tasks, compared to the results of sampling more than one clients. This supports that our sampling strategy is an efficient way to train the Transformer body with less computational cost even when the number of clients increases.

Table 9: Results of the study on the sampling strategy of clients in task-agnostic learning.

| Task | Deblocking | | | | Denoising | | Deraining | | | | Deblurring | |
| | BSDS500 (Q10) | | BSDS500 (Q50) | | CBSD68 ($\sigma = 30$) | | Rain100H | | Rain100L | | GoPro | |
| Sampling | PNSR | SSIM | PNSR | SSIM | PNSR | SSIM | PNSR | SSIM | PNSR | SSIM | PNSR | SSIM |
|---|---|---|---|---|---|---|---|---|---|---|---|---|
| 1 | 27.57 | 0.782 | 33.01 | 0.922 | 30.62 | 0.869 | 28.75 | 0.862 | 32.69 | 0.936 | 29.09 | 0.873 |
| 2 | 27.57 | 0.782 | 33.01 | 0.922 | 30.62 | 0.869 | 28.68 | 0.861 | 32.59 | 0.935 | 29.10 | 0.873 |
| 4 | 27.57 | 0.782 | 33.01 | 0.922 | 30.62 | 0.869 | 28.75 | 0.862 | 32.35 | 0.933 | 29.11 | 0.873 |

### D.4  COMMUNICATION COST BETWEEN CLIENTS AND SERVER

In the proposed TAViT, the features and gradients of the networks are transported between clients and server, so one may wonder how much additional communication cost occurs. To compute the communication cost of our method, we assume that the cost is proportional to the number of transported elements. Also, since the size of features and gradients from clients to the server are the same with those from the server to client, we only consider one direction from clients to the server. Then, we computed the maximum cost for one communication to update our model for each task-specific and task-agnostic learning, and compared our cost to the method of FL (McMahan et al., 2017a).

Specifically, when there $N_k$ clients for the $k$-th task, let $P_H$, $P_B$ and $P_T$ be the number of parameters of the head, body, and tail, respectively. In the case of FL that aggregates the whole model composed of the head, body, and tail, the communication cost per communication can be represented as:

$$Cost_{FL} = N_k(P_H + P_B + P_T). \tag{13}$$

On the other hand, our model does not require the transportation of learnable parameters except for the aggregation step in the task-specific learning. Thus, the communication cost can be computed as follows:

$$Cost_{TAViT} = \begin{cases} N_k(P_H + P_T), & \text{if an aggregation step} & \text{(task-specific learning)} \\ N_k(F + G), & \text{else if a non-aggregation step} & \text{(task-specific learning)} \\ F + G, & \text{otherwise}, & \text{(task-agnostic learning)} \end{cases} \quad (14)$$

where $F$ and $G$ are the number of elements of transported features and gradients, respectively.

From (14), we can see that the communication cost at the network aggregation step in the task-specific learning of the proposed method is smaller than FL that needs to transport the parameters of the whole model including head, body, and tail. Specifically, instead of aggregating parameters of the Transformer body, the TAViT transports features and gradients that have much smaller size than the body parameters, which can reduce the cost per communication significantly. For example, the proposed model for the deblocking task contains $P_H + P_B + P_T = 44,774,792$ parameters whose memory size is about 207.5MB. Suppose that 10 clients participate in FL to train this model. Then, 447.7M elements are transported from the clients to the server, and the network of the server should handle more than 2GB load per communication. In contrast, our model transports $P_H + PT = 27,952,520$ parameters whose memory size is approximately 142.5MB. Thus, even with 10 clients, 279.5M elements are transported, and the network of the server is supposed to handle about 60% load of FL. In addition, since the number of features and gradients is $F = G = 20 \times 16 \times 16 \times 512 = 2,621,440$ which is 10MB of memory, the number of transported elements per communication for 10 clients is 52.4M, and the server is pressed by only 200MB load per communication.

On the other hand, in task-agnostic learning, the server updates the body with the sampled client without any weight aggregation. Accordingly, only the features and gradients are transported from the client to the server. In particular, in the case of the communication from the server to the client, the server does not need to transport the gradients to the client, but only transmit the features. Thus, the cost per communication in the task-agnostic learning phase is significantly reduced.

Therefore, up to a certain epoch size, our model is more communication bandwidth efficient compared to the classical FL, and the advantage increases more if a bigger Transformer body is used for better representation of global attention.

**Scalability** Suppose that there are $K$ tasks, and let the total number of clients connected to a server be $N_{all}$. For simple analysis of the scalability, we assume that each communication between clients and the server takes a constant time. The scalability is computed on the time complexity for one communication between clients and the server to update the models. For our task-specific learning, the one communication has time complexity $O(N_{all})$ if we update the heads and tails of all clients. This means that the communication cost would increase according to the number of clients, which can limit the scalability of the proposed method. However, if we apply the client sampling strategy of the FedAvg, we can control the number of communications and the one communication will have time complexity $\Omega(K)$. This sampling strategy can be readily adapted to our model without significant modification. On the other hand, for the task-agnostic learning phase, the one communication has time complexity $O(K)$, since the network parameters of clients for the same task are aggregated before the task-agnostic optimization. Also, according to the sampling strategy of one task in the proposed method, one communication has time complexity $\Omega(1)$, which is studied in Appendix D.3.

## D.5 APPLICATION TO HIGH-LEVEL VISION TASKS AND MEDICAL DATA

In the main paper, the proposed TAViT was demonstrated on multiple low-level computer vision tasks. However, the TAViT framework can be also used for a wide range of high-level computer vision tasks, and even with different data domains such as medical images. To demonstrate this, we additionally conduct on inpainting task for natural images and denoising task for X-ray CT images. Here, the image inpainting is a higher-level computer vision task which requires more semantic information, and the denoising of X-ray CT requires domain-specific knowledge about the data. In particular, to show that our task-agnostic Transformer body provides a positive effect on the training of new task-specific networks, we performed the task-specific learning only for the client-side heads and tails by subscribing to the pre-trained Transformer body, which was trained on the four natural image processing tasks without additional fine-tuning. The details of training and results are as follows.

**Dataset**    For the image inpainting task, we used PASCAL VOC2012 dataset which contains 10,582 natural images. The information about license and source of this dataset can be found in Appendix B. For the preprocessing, we scaled the image from [0, 255] to [-1, 1] and randomly cropped by $128 \times 128$ patches. Then we multiplied the image with the zero-box mask that has a random size of width and height from 48 to 64 according to Yu et al. (2018). For the X-ray CT denoising task, we used the 2016 AAPM Low-dose CT Grand Challenge dataset (McCollough et al., 2020) that provides noisy CT images with quarter dose and clean CT images with routine dose of X-ray. Since the X-ray CT data are measured in the Hounsfield unit, we divided the intensity by 4,000 and randomly cropped by $64 \times 64$ patches.

**Implementation details**    For the image inpainting task, we employed the network architecture of Yu et al. (2018) and decomposed it into two parts for the task-specific head and tail. We performed task-specific learning by minimizing the adversarial generative loss for 400 epochs using Adam optimizer with learning rate $1 \times 10^{-4}$. For the X-ray CT denoising task, we used the same network architecture of head and tail implemented in this paper. We trained the task-specific networks with the fixed task-agnostic body for 400 epochs using Adam optimization algorithm with learning rate $5 \times 10^{-3}$.

**Results**    To evaluate the performance of image inpainting and medical image denoising, we compared our method to the CNN model that has the same network architectures of head and tail with ours but does not have the Transformer body. The quantitative evaluation results are shown in Table 10, and the visual comparisons are shown in Figure 13. We can see that the performance of inpainting is improved when training the client-side networks with our pre-trained Transformer body, even though we use the Transformer body pre-trained using low-level computer vision tasks. This implies that the proposed method can be extended to various high-level tasks. In addition, we can observe that our model on the medical image denoising task achieves higher performance than the comparative CNN model and provides clean images, although the Transformer body was trained on the natural image domain. From these results, we can confirm that our task-agnostic Transformer body has a capability to learn the domain gap even in different data sources. Also, this suggests that clients do not need to train the server-side body from the scratch when they subscribe the body for the other tasks.

Table 10: Results of applications to diverse image processing tasks using different image domains.

| Task | Domain | CNN | | TAViT | |
|------|--------|-----|-----|-------|-----|
| | | PSNR | SSIM | PSNR | SSIM |
| Inpainting | Natural Image | 24.50 | 0.861 | **24.85** | **0.865** |
| Denoising | Medical Image (CT) | 41.28 | 0.958 | **42.05** | **0.961** |

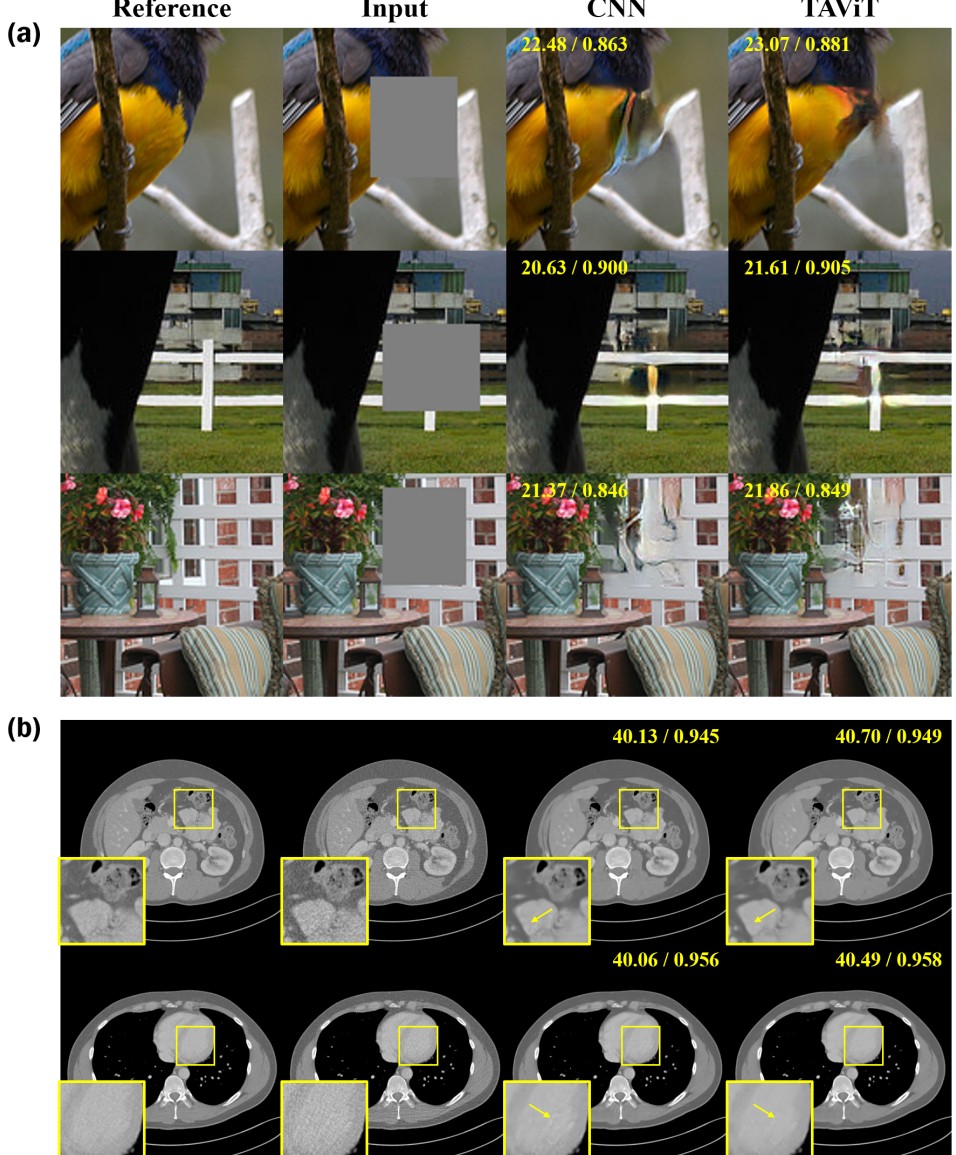

Figure 13: Visual comparisons of the application results: (a) inpainting of natural images, and (b) denoising of medical CT images. The average values of PSNR / SSIM are displayed on each result.

