# OpenReview forum: "Privacy-preserving Task-Agnostic Vision Transformer for Image Processing"
_ICLR.cc/2022/Conference — ICLR 2022 Submitted_

### Official Review · Reviewer_iupq · 2021-10-19

**Correctness:** 3
**Technical Novelty And Significance:** 3
**Empirical Novelty And Significance:** 4
**Recommendation:** 8
**Confidence:** 3

**Main Review:**

# Strengths
* It gives a new and practical distributed learning framework for image restoration tasks. It is capable of handling multiple tasks with maintaining privacy.
* It gives state-of-the-art or competitive results on the evaluated restoration tasks.
* The paper is easy to follow.

# Weaknesses
* The emphasized privacy-preserving property of the given framework is not experimentally or theoretically validated.
* No explanation about why only choosing deblocking, denoising, deraining, and deblurring in clients. What about super-resolution and inpainting?

**Summary Of The Paper:**

This paper presents a new distributed learning framework exploiting the vision transformer for various image processing applications. It gives impressive quantitative and qualitative results on multiple image restoration tasks meanwhile keeping privacy. Specifically, it employs a task-agnostic vision transformer to learn universal representation at the server, and several CNN-based task-specific heads and tails to handle different image restoration tasks at the client side. It also gives a training strategy to learn this model.

**Summary Of The Review:**

I vote for accepting this paper. Its technical novelties and contributions are sufficient, and the given system seems practical and effective. It applies federal learning to image restoration tasks. It leverages ViT for universal representation learning, and task-specific heads and tails for training different tasks. The results are convincing. I think it is worth giving a brief study about how this framework reacts to privacy attacks.

---

> ### Author Response · Authors · 2021-11-21
> **Response to reviewer iupq**
>
> $#$**C1. The emphasized privacy-preserving property of the given framework is not experimentally or theoretically validated.**
>
> Thank you for the valuable comment. As your suggestion, we discussed the privacy-preserving property of the proposed method in Section 3.3 and Appendix D.1.
>
> In fact, another powerful and unique mechanism for maintaining privacy in our method arises when the proposed method's client-side network has a skip connection between the head and the tail. In this case, the transported feature characteristic can contain very lossy information from the original data. As shown in Appendix D.1, one cannot reconstruct data only using the transmitted hidden features of the proposed method.
>
> Nevertheless, since this work is more like a proof-of-the-concept study, rather than a ready-to-use solution for the industry, the privacy-related issues such as threatening privacy via inversion attack should be further analyzed for practical application.
>
> $#$**C2. No explanation about why only choosing deblocking, denoising, deraining, and deblurring in clients. What about super-resolution and inpainting?**
>
> Thanks for your constructive comment. Although we chose those tasks to demonstrate our method experimentally, our model can be also applied to various image processing tasks according to the configuration of the head and tail networks of clients.
>
> To show this, we conducted an additional experiment on image inpainting, which is considered a high-level computer vision task. Specifically, by subscribing to our pre-trained Transformer body for low-level computer vision tasks, we designed CNN-based head and tail that can be connected to the body. Here, for image inpainting, we employed the network architecture of [1]. Then, we trained only the client-side head and tail for the inpainting in a task-specific learning manner by freezing the Transformer body. To evaluate the inpainting performance, we compared our method to the model composed of the same CNN head and tail without the Transformer body. As shown in below Table 1, the performance of inpainting is improved when training the client-side networks with our pre-trained Transformer body. This implies that the proposed method can be extended to various tasks. We discussed this in more detail in Appendix D.5.
>
> **Table 1.** Results of applications to image inpainting tasks.  The average values of PSNR / SSIM are displayed.
>
> |     Task    |        CNN        |        Ours       |
> |:-----------------------:|:------------------:|:-----------------:|
> |       Inpainting        | 24.50 dB / 0.861 | **24.85 dB / 0.865** |
>
> [1] Yu, Jiahui, et al. "Generative image inpainting with contextual attention." Proceedings of the IEEE conference on computer vision and pattern recognition. 2018.

---

### Official Review · Reviewer_QjZj · 2021-10-22

**Correctness:** 3
**Technical Novelty And Significance:** 2
**Empirical Novelty And Significance:** 2
**Recommendation:** 6
**Confidence:** 3

**Main Review:**

- In line-5 of the abstract, what did the authors inspire from Vision Transformers? Why the core motivation of this paper comes from "the success of ViT". The authors should revise the statement.
- The proposed method has been verified on four low-level tasks. Can this strategy/framework be applied to higher-level tasks, such as image inpainting, classification, and object detection? Note that those tasks need more semantic understanding during training/inference.
- In the traditional federated learning framework, each client commonly conducts the local train process, then transfers the model update consisting of the intermediate gradient to sever. However, in this paper, the author transfers the dataset's features to sever, which may bring about unpredictable challenges. For example, the computation cost of homomorphic encryption for features may increase rapidly. Does it have any advantages (research or application value)?
- As for different tasks, the authors use a unified task-agnostic Transformer body, how the authors bridge the domain gap caused by the other tasks' knowledge. Furthermore, do different data sources (e.g., nature, satellite, and medical images) share the same Transformer body in Figure-1? Does it still work well? The authors should provide more experimental results and in-depth analysis to verify this point.
- Prior works  [*1,*2] have also addressed the task-agnostic problem in federated learning. What's the significant difference between those works?

[*1] Federated Learning with Unbiased Gradient Aggregation and Controllable Meta Updating
[*2] Task-Agnostic Privacy-Preserving Representation Learning via Federated Learning

**Summary Of The Paper:**

In this work, the authors present a multi-task distributed learning framework called TAViT. The task-specific head CNN and the tail CNN are distributed to clients with their data connected to a standard Transformer body placed in the server. With an alternating training scheme, the heads and tails on client sides are trained by task-specific learning, while the body is trained by task-agnostic learning. Experiments on four different image processing tasks show the success of task-agnostic learning of the Transformer body and its synergistic improvement with the task-specific heads and tails.

**Summary Of The Review:**

This paper is well-written and presented. However, some key experiments and designs confuse me a lot. Overview, it approaches the borderline of the ICLR community. The authors should address the above concerns (# Main Review) in the rebuttal period.

After reading the responses from the authors, the authors partially solved my concerns. Thus, I decided to increase the rating.

---

> ### Author Response · Authors · 2021-11-21
> **Response to reviewer QjZj (5/5)**
>
> $#$**C5. Prior works [1,2] have also addressed the task-agnostic problem in federated learning. What's the significant difference between those works?**
>
> **[1] Federated Learning with Unbiased Gradient Aggregation and Controllable Meta Updating**
>
> **[2] Task-Agnostic Privacy-Preserving Representation Learning via Federated Learning**
>
> The first work [*1] is a FL method on non-IID data. This paper presents a local gradient tracking algorithm and meta updating using metadata on the server to reduce the undesired effect from the gap of local data distribution. The main differences between this work and our method is as follows.
>   - While the client-side networks of the prior work require the same architecture to be aggregated on the server, those of our method can have different structures for different tasks.
>   - In contrast to the prior work that aggregates the whole network, the Transformer body of our method is shared by clients and not aggregated.
>   - The prior work requires additional metadata on the server for task-agnostic learning, but our model uses client-side distributed data in both task-specific and task-agnostic learning.
> - The prior work designs a method for a single task with distributed non-IID data, whereas our TAViT handles multiple image processing tasks.
>
> The second work [*2] proposes a framework of TAP that consists of a feature extractor, mutual information estimator, and adversarial classifier. In particular, the feature extractor is trained not to convey private attributes but to retain as much information as possible for downstream tasks, through the estimator and adversarial classifier. Then, clients train task-specific classifiers using the feature extractor. The major differences are listed as follows:
>   - In contrast to the previous work that uses the common feature extractor aggregated for sampled clients, our task-agnostic body that can be considered as a feature extractor is a single model on the server so that it is not trained by FL. The task-specific heads for different tasks of our model need not be synchronized.
>   - While the downstream classifiers in TAP take features extracted only from the common network, our task-specific tails of TAViT take features from both heads and the task-agnostic body so that the tails can generate outputs using the common features as well as task-specific features, which enables our model to learn multiple distinct tasks using local data.
>   - Each downstream classifier in the prior work is trained independently based on the feature extractor, whereas each task-specific head and tail of our TAViT is jointly trained with a common body, which improves the performance further through the alternative learning scheme.
> We have included the discussion on the above two works in Section 2 of the main paper.

---

> ### Author Response · Authors · 2021-11-21
> **Response to reviewer QjZj (4/5)**
>
> $#$**C4. As for different tasks, the authors use a unified task-agnostic Transformer body, how the authors bridge the domain gap caused by the other tasks' knowledge. Furthermore, do different data sources (e.g., nature, satellite, and medical images) share the same Transformer body in Figure-1? Does it still work well? The authors should provide more experimental results and in-depth analysis to verify this point.**
>
> As described in Section 1 and 3, we bridge the domain gap between different tasks by the Transformer body that can learn the common representation regardless of different features from the task-specific heads. We demonstrate this through the experimental results. In particular, Figure 3 in the main paper implies that the Transformer body plays a role of not only bridging the domain gap but also giving a synergy effect to the task-specific heads and tails. Specifically, with each alternating learning cycle, we can observe the performance improvements for most tasks. Although the deraining performance on the Rain100L dataset decreases slightly due to the domain gap as you mentioned, our model shows better results as the alternating learning continues. This supports the effectiveness of using the task-agnostic Transformer body.
>
> Per your suggestion, we also performed additional study on whether the task-agnostic body for natural image processing can be applied to different data sources. Specifically, for the denoising task on medical images, we conducted the task-specific learning for the client-side head and tail using CT images by subscribing to the Transformer body trained on natural images. (Here, we used the AAPM dataset that provides noisy CT images with quarter dose and clean CT images with routine dose of X-ray. We added a description of the dataset and data processing in Appendix D.5). When we compared the performance with CNN model that is trained without the body, as reported in below Table 2, our model shows higher PSNR and SSIM and provides clean images (See Appendix D.5).  From these results, we can confirm that our task-agnostic Transformer body has a capability to learn the domain gap even in different data sources.
>
> **Table 2.** Results of applications to the denoising of CT image. The average values of PSNR / SSIM are displayed.
>
> |       Task (domain)        |       CNN        |       Ours       |
> |:----------------:|:----------------:|:----------------:|
> | Denoising (medical image) | 41.28 dB / 0.958 | 42.05 dB / 0.961 |

---

> ### Author Response · Authors · 2021-11-21
> **Response to reviewer QjZj (3/5)**
>
> $#$**C3. In the traditional federated learning framework, each client commonly conducts the local train process, then transfers the model update consisting of the intermediate gradient to sever. However, in this paper, the author transfers the dataset's features to sever, which may bring about unpredictable challenges. For example, the computation cost of homomorphic encryption for features may increase rapidly. Does it have any advantages (research or application value)?**
>
> Thank you for the valuable comment. The advantages of our method are as follows, which are discussed in the Section 3.3 and Appendix D.4.
>
> Specifically, in the view of the cost per communication, our method has an advantage over traditional FL, since our model alternates parameter aggregation of the large Transformer body by much smaller features and gradients. In our experiments, the proposed method requires the amount of elements to be transported about 60% of the FL. Also, the size of features and gradients were 6.5 times smaller than the Transformer body. In addition, the sampling strategy for task-agnostic learning can control the communication cost, and we can also readily apply the client sampling strategy of FedAvg for task-specific learning, which further reduces communication costs. Therefore,  up to a certain epoch size,  our model is more communication bandwidth efficient compared to the classical FL, and the advantage increases more if a bigger Transformer body is used for better representation of global attention.
>
> Furthermore, our method has a strong merit that we present a distributed learning method for distinct tasks. In contrast to the existing FL methods that all clients should have the same network architecture to share the parameters, the participating clients of our model can design and train their own networks depending on the task using local data. Also, they can achieve comparable or higher performance by subscribing to a task-agnostic Transformer body without fine-tuning, as shown in our experimental results in Section D.5 of Appendix. This can be usefully applied when clients want to not only train the networks without sharing data but also achieve high-quality outputs, such as in medical fields where sharing patient data with other institutions is sensitive.

---

> ### Author Response · Authors · 2021-11-21
> **Response to reviewer QjZj (1-2/5)**
>
> $#$**C1. In line-5 of the abstract, what did the authors inspire from Vision Transformers? Why the core motivation of this paper comes from "the success of ViT". The authors should revise the statement.**
>
> The reason for developing our model with CNN-based heads / tails and the Transformer-based body is to take advantage of each network. In particular, Transformer learns the global attention of the input sequence through self-attention modules and has recently been extensively studied for various computer vision tasks. One of the most unique advantages of Transformer is to convert "unattended'' input feature vectors into "attended" output feature vectors by learning global attention and non-local interactions between the input features. Accordingly, the task-specific head / tail can be only trained to learn task-specific local features, whereas the global feature can be learned through the Transformer body. This disentangled representation of local and non-local features has been pursued throughout the development of deep networks. Thus, the proposed Transformer-based approach is considered to be one of the most advanced architectures for achieving this goal, as it improves synergistically the overall performance, as shown in our experiments, and at the same time leads the privacy-preserving split-learning architecture. We added this motivation in the Abstract and Introduction.
>
> $#$**C2. The proposed method has been verified on four low-level tasks. Can this strategy/framework be applied to higher-level tasks, such as image inpainting, classification, and object detection? Note that those tasks need more semantic understanding during training/inference.**
>
> Thanks for your constructive comment. Although we demonstrated our model on the low-level image processing tasks, the proposed method can be also applied to high-level computer vision tasks by choosing appropriate head and tail networks.
>
> To demonstrate this, we conducted an additional experiment on image inpainting per your suggestion. Specifically, by using the pre-trained Transformer body for natural image processing, we designed CNN-based head and tail that can be connected to the body. Here, for image inpainting, we employed the network architecture of [1]. Then, we trained only the client-side head and tail for the inpainting in a task-specific learning manner by freezing the Transformer body. To evaluate the effect on the high-level task, we compared our method to the model with the same CNN architecture without the Transformer body. As shown in Table 1, the performance of inpainting is improved when training the client-side networks with our pre-trained Transformer body, even though the body was trained on low-level vision tasks. This implies that the proposed method can be extended to various high-level computer vision tasks, and also clients do not need to train the server-side body from the scratch when they use the body for other tasks. We discussed this in more detail in Appendix D.5.
>
> **Table 1.** Results of applications to image inpainting tasks.  The average values of PSNR / SSIM are displayed.
>
> |     Task (domain)     |        CNN        |        Ours       |
> |:-----------------------:|:------------------:|:-----------------:|
> |       Inpainting  (natural image)      | 24.50 dB / 0.861 | **24.85 dB / 0.865** |
>
> [1] Yu, Jiahui, et al. "Generative image inpainting with contextual attention." Proceedings of the IEEE conference on computer vision and pattern recognition. 2018.

---

### Official Review · Reviewer_kCVH · 2021-10-31

**Correctness:** 4
**Technical Novelty And Significance:** 2
**Empirical Novelty And Significance:** 2
**Recommendation:** 6
**Confidence:** 3

**Details Of Ethics Concerns:**

There is no ethical concern per se. As mentioned above, it's just that the ethics statement contains a discussion that should have been in the main paper (and greatly expanded on there). Also, the title ("privacy-preserving") might overstate the guarantees the presented architecture can give.

**Main Review:**

The submission is overall clearly written and presents an embodiment of a split architecture between client(s) and server that facilitates multi-task learning, which could be adopted by further work in the future. Code and models are available, which greatly eases adoption.

However, while most of the pages are spent on architecture description and experimental results, there are several key omissions of discussion points which I deem essential for a paper in the distributed learning space:
- There is no mention of any sort of privacy guarantees given the proposed architecture, despite "privacy-preserving" being part of the title. Privacy preservation is a strong claim that needs to be backed up rigorously. Moreover, in their ethics statement, the authors correctly lay out that transmitted hidden features "may leak the raw data to some degree."
- There is no discussion of communication cost in a federated setting. The number of clients used in experiments is exceedingly small, which might serve as a proof of concept. But given that gradients have to be transmitted two-way or one-way for training head/tail and body parts of the architecture, a discussion on communication cost and scalability is necessary.
- Several key choices are not sufficiently motivated, including the choice of both CNN and transformer architectures, or the sampling of exactly one client from each task in Eq. 5. What makes CNN architectures less suitable for the body part in a more general framework beyond the particular architecture embodiment presented here? How would the presented sampling strategy scale in the face of several orders of magnitudes of more clients?

If the focus of the paper is to position TAviT as a general distributed multi-task learning framework, then the diversity of presented experiments for validation could have been expanded. On the other hand, if the focus is to present this particular architecture as a viable means to do distributed multi-task learning for the presented tasks, then the results of Table 2 remain unconvincing, in the sense that differences in results might come from essentially uncomparable architectures, as opposed to contributions to multi-task learning or distributed learning.

--------------------
Post-rebuttal:
I thank the authors for their valuable comments on my review and their revision. The revision has improved the submission substantially. My comments were addressed mainly due to the addition of Section 3.3 and the there referenced Appendix D. I believe that most of the other reviewers' comments were also addressed and can now recommend the submission for acceptance. This is reflected by my adjusted score.

Minor comment:
- The "Federated Learning" paragraph on pg. 3 contains an unresolved reference due to a typo.




**Summary Of The Paper:**

The paper presents an architecture for image processing tasks that splits up a network into three subsequent parts: head, body, and tail. Head and tail parts are CNN-based and can be trained on multiple client devices using federated learning (FedAvg), while the body part of the architecture is transformer-based and is trained on a central server. Head and tail parts are trained for specific tasks, while the body part is trained in a task-agnostic manner by selecting clients from each task for loss optimization.
Experimental results show benchmark and convergence results that are comparable or favorable to non-distributed models, as well as comparison results to purely FL and SL approaches with a very small nr of clients.

**Summary Of The Review:**

The submission presents a specific system that combines split and federated learning for multi-task learning of various image processing applications. It offers a good proof of concept of the proposed architecture decomposition, but lacks a robust discussion on communication cost/overhead as well as privacy guarantees. In particular, the ethics statement relativizes what the title claims ("privacy-preserving").

---

> ### Author Response · Authors · 2021-11-21
> **Response to reviewer kCVH (4/4)**
>
> $#$**C4. If the focus of the paper is to position TAViT as a general distributed multi-task learning framework, then the diversity of presented experiments for validation could have been expanded. On the other hand, if the focus is to present this particular architecture as a viable means to do distributed multi-task learning for the presented tasks, then the results of Table 2 remain unconvincing, in the sense that differences in results might come from essentially uncomparable architectures, as opposed to contributions to multi-task learning or distributed learning.**
>
> Thanks for your valuable comments. To demonstrate our TAViT can be applied to diverse image processing tasks as well as the presented experiments in the main paper, we additionally conducted the experiment on inpainting of natural images and denoising of medical images. In particular, to show that our task-agnostic Transformer body provides a positive effect on the training of new task-specific networks, we performed the task-specific learning by freezing the body that was trained with the presented four tasks. As shown in below Table 1, when comparing to the CNN model that does not have the Transformer body but has the same head and tail with ours, we can see that the proposed Transformer body plays a role to process the common feature representation and provide higher performance for various tasks and image domains. (See Appendix D.5.)
>
> In regard to your comment, the reason we display the results of CNN-based methods for each task is for one to compare how high the performance our method can provide.  On the other hand, when we compared our method to the other distributed learning such as SL and FL (Table 1 in the main paper), end-to-end learning and the single-task learning under the distributed data (Table 2 in the main paper), we performed experiments with the same conditions for the network architectures and the training.
>
> **Table 1.** Results of applications to diverse image processing tasks using different image domains. The average values of PSNR / SSIM are displayed.
>
> |     Task (domain)     |        CNN        |        Ours       |
> |:----------------:|:-----------------:|:-----------------:|
> | Inpainting (Natural image) | 24.50 dB / 0.861 | **24.85 dB / 0.865** |
> |  Denoising (Medical image, CT)  | 41.28 dB / 0.958 | **42.05 dB / 0.961** |

---

> ### Author Response · Authors · 2021-11-21
> **Response to reviewer kCVH (3/4)**
>
> $#$**C3. Several key choices are not sufficiently motivated, including the choice of both CNN and transformer architectures, or the sampling of exactly one client from each task in Eq. 5. What makes CNN architectures less suitable for the body part in a more general framework beyond the particular architecture embodiment presented here? How would the presented sampling strategy scale in the face of several orders of magnitudes of more clients?**
>
> **[Motivation of the choice of model architectures]**
>
> The reason for developing our model with CNN-based heads / tails and the Transformer-based body is to take advantage of each network. In particular, Transformer learns the global attention of the input sequence through self-attention modules and has recently been extensively studied for various computer vision tasks. One of the most unique advantages of the Transformer body is to convert "unattended'' input feature vectors into "attended" output feature vectors by learning global attention and non-local interactions between the input features. Accordingly, the task-specific head / tail can only be trained to learn task-specific local features, whereas the global feature can be learned through the Transformer body. This disentangled representation of local and non-local features has been pursued throughout the development of deep networks [1,2,3], and the proposed Transformer-based approach is considered to be one of the most advanced architectures for achieving this goal, as they improve synergistically the overall performance, as shown in our experiments, and at the same time lead the privacy-preserving split-learning architecture.
>
> To verify the synergistic effects of the Transformer body that is suitable for distributed learning, we performed an additional experiment by replacing the Transformer body with a CNN model.  In particular, we configured the CNN body with C-B-R blocks, where C is a convolution layer, B is a batch normalization layer, and R is a ReLu layer. For a fair comparison, we've set the number of C-B-R blocks so that they have almost the same number of learnable parameters as the Transformer body we are using. Then, using this CNN body, we implemented our proposed task-specific and task-agnostic learning for the multiple image processing tasks similar to our method in the main paper. As a result, Table 1 (below) shows that our method with the Transformer body achieves higher performance in distributed learning with multiple tasks compared to the model with the CNN body. We added this study to Appendix D.2.
>
> Furthermore, we have also applied our model to additional natural image inpainting and medical image denoising tasks. By fixing the Transformer body, which was pre-trained for the four tasks in the main paper, we only trained the task-specific heads and tails on clients. Even without fine-tuning the Transformer body, our task-specific networks offer better image quality compared to the model that consists of just a CNN-based encoder and decoder. This implies that the Transformer body plays a significant role in learning a common feature representation for different tasks and also provides sufficient capacity for image processing of different image domains such as medical images. See Appendix D.5.
>
> [1] Ye, J.C., Han, Y. and Cha, E., 2018. Deep convolutional framelets: A general deep learning framework for inverse problems. SIAM Journal on Imaging Sciences, 11(2), pp.991-1048.
>
> [2] Zhang, S., He, X. and Yan, S., 2019, May. Latentgnn: Learning efficient non-local relations for visual recognition. In International Conference on Machine Learning (pp. 7374-7383). PMLR.
>
> [3] Wang, X., Girshick, R., Gupta, A. and He, K., 2018. Non-local neural networks. In Proceedings of the IEEE conference on computer vision and pattern recognition (pp. 7794-7803).
>
> **[Sampling strategy of TAViT]**
>
> In task-agnostic learning, one client is sampled for each iteration. Here, since the networks of clients for the same task are aggregated before the task-agnostic learning, we can readily sample one client for each task. Then, choosing one client for the subset of Eq(5) can be viewed as sampling one task, which reduces communication cost naturally. In fact, the performance of TAViT is not affected by the number of sampled clients for task-agnostic learning, since the task-agnostic body is updated for sufficient iterations. To demonstrate this, we performed the task-agnostic learning for the four tasks in our experiments by varying the number of sampled clients. As shown in Table 2, sampling one client shows comparable or higher performance, compared to sampling two or all clients. This supports that our sampling strategy is an efficient way to train the Transformer body with less communication cost even when the number of clients increases. We added this in Appendix D.3.

---

> > ### Author Response · Authors · 2021-11-21
> > **Response to reviewer kCVH (3/4): Tables**
> >
> > **Table 1.** Results of study on the effect of the Transformer body of TAViT. The average values of PSNR / SSIM are displayed.
> >
> > |       Task      |       Input       |  Cycle 0.5 (CNN body) |  Cycle 1.0 (CNN body) | Cycle 0.5 (Transformer body) | Cycle 1.0 (Transformer body) |
> > |:---------------:|:-----------------:|:----------------:|:----------------:|:-----------------------:|:-----------------------:|
> > | Deblocking (Q10) |  25.67 dB / 0.719 | 26.42 dB / 0.752 | 26.49 dB / 0.755 |     27.53 dB / 0.781    |    **27.57 dB / 0.782**    |
> > | Deblocking (Q50) | 31.51 dB / 0.902 | 32.92 dB / 0.921 | 32.95 dB / 0.921 |     32.92 dB / 0.921    |    **33.01 dB / 0.922**    |
> > |     Denoising    |  19.03 dB / 0.336 | 30.22 dB / 0.866 | 30.50 dB / 0.866 |    30.57 dB / 0.868    |    **30.62 dB / 0.869**    |
> > |  Deraining (RainH) | 13.55 dB / 0.380 | 28.23 dB / 0.851 | 28.26 dB / 0.852 |    28.24 dB / 0.860    |    **28.75 dB / 0.862**    |
> > |  Deraining (RainL) | 26.90 dB / 0.839 | 31.58 dB / 0.930 | 31.77 dB / 0.930 |    **33.17 dB / 0.939**    |    32.69 dB / 0.936    |
> > |      Deblurring     |  25.64 dB / 0.790 | 28.78 dB / 0.865 | 28.94 dB / 0.868 |    28.94 dB / 0.871    |    **29.09 dB / 0.873**    |
> >
> > **Table 2.** Results of the study on the sampling strategy in task-agnostic learning. The average values of PSNR / SSIM are displayed.
> >
> > |                    | 1 client sampling  | 2 clients sampling | 4 clients sampling  |
> > |:------------------:|:------------------:|:------------------:|:-------------------:|
> > | Deblocking (Q10)  |  27.57 dB / 0.782  |  27.57 dB / 0.782  | 27.57 dB / 0.782  |
> > | Deblocking  (Q50) |  33.01 dB / 0.922  |  33.01 dB / 0.922  |  33.01 dB / 0.922   |
> > |      Denoising     |  30.62 dB / 0.869  |  30.62 dB / 0.869  |  30.62 dB / 0.869   |
> > | Deraining (RainH)  |  28.75 dB / 0.862  |  28.68 dB / 0.861  |  28.75 dB / 0.862   |
> > | Deraining (RainL)  |  32.69 dB / 0.936  |  32.59 dB / 0.935  |  32.35 dB / 0.933   |
> > |      Deblurring    |  29.09 dB / 0.873  |  29.10 dB / 0.873  |  29.11 dB / 0.873  |

---

> ### Author Response · Authors · 2021-11-21
> **Response to reviewer kCVH (2/4)**
>
> $#$**C2. There is no discussion of communication cost in a federated setting. The number of clients used in experiments is exceedingly small, which might serve as a proof of concept. But given that gradients have to be transmitted two-way or one-way for training head/tail and body parts of the architecture, a discussion on communication cost and scalability is necessary.**
>
> Thank you for the constructive comment. According to your suggestion, we have now computed the communication cost of our model and discussed scalability when the number of clients increases as follows. Here, the cost and scalability are computed in the viewpoint of clients. Please refer to Appendix D.4.
>
> **[Communication cost]**
>
> We assume that the communication cost is proportional to the number of transported elements. Also, since the size of features and gradients from clients to the server are the same as those from the server to client, we only consider one direction from clients to the server. The transported elements from clients to the server are as follows:
>
> - $P_{H}$: the number of parameters of the head
> - $P_{B}$: the number of parameters of the body
> - $P_{T}$: the number of parameters of the tail
> - $F$: the number of elements of transported features
> - $G$: the number of elements of transported gradients
>
> Then, for $N_k$ clients, we can write the cost per communication based on the number of transported elements as:
> - FL (FedAvg): $ N_{k}(P_{H} + P_{B} + P_{T})$
> - Task-specific learning of TAViT at aggregation step: $ N_{k}(P_{H} + P_{T})$
> - Task-specific learning of TAViT at non-aggregation step: $ N_{k}(F+G)$
> - Task-agnostic learning of TAViT: $ F+G$
>
> From the above table, we can see that the communication cost at the network aggregation step in our task-specific learning step is smaller than the standard federated learning (FL) that needs to transport the parameters of the whole model. Specifically, instead of aggregating parameters of the Transformer body, the TAViT transports features and gradients that have a much smaller size than the body parameters, which can reduce the cost per communication significantly.  For example, the proposed model for the deblocking task contains $P_{H} + P_{B} + P_{T}=44,774,792$ parameters whose memory size is about 207.5MB. Suppose that 10 clients participate in the classical FL to train this model. Then, 447.7M elements are transported from the clients to the server, and the network of the server should handle more than 2GB load per communication. In contrast, our model transports $P_{H}+P_{T}=27,952,520$ parameters whose memory size is approximately 142.5MB. Thus, even with 10 clients, 279.5M elements are transported, and the network of the server is supposed to handle about 60% load of FL. In addition, since the number of features and gradients is $F=G=20\times 16\times 16\times 512=2,621,440$ which is 10MB of memory, the number of transported elements per communication for 10 clients is 52.4M, and the server is pressed by only 200MB load per communication.
>
> On the other hand, in task-agnostic learning, the server updates the body with the sampled client without any weight aggregation. Accordingly, only the features and gradients are transported from the client to the server. In particular, in the case of the communication from the server to the client, the server does not need to transport the gradients to the client, but only transmits the features. Thus, the cost per communication in the task-agnostic learning phase is significantly reduced.
>
> Therefore,  up to a certain epoch size,  our model is more communication bandwidth-efficient compared to the classical FL, and the advantage increases more if a bigger Transformer body is used for better representation of global attention.

---

> > ### Author Response · Authors · 2021-11-21
> > **Response to reviewer kCVH (2/4)**
> >
> > **[Scalability]**
> >
> > Suppose that there are ${K}$ tasks, and let the total number of clients connected to a server be $N_{all}$. For a simple analysis of the scalability, we assume that each communication between clients and the server takes a constant time. The scalability is computed on the time complexity for one communication between clients and the server to update the models.
> >
> > For our task-specific learning, the one communication has time complexity $O(N_{all})$ if we update the heads and tails of all clients. This means that the communication cost would increase according to the number of clients, which can limit the scalability of the proposed method. However, if we apply the client sampling strategy similar to the FedAvg, we can control the number of communications and the one communication will have time complexity $\Omega (K)$. This sampling strategy can be readily adapted to our model without significant modification.
> >
> > For the task-agnostic learning phase, the one communication has time complexity $O(K)$, since the network parameters of clients for the same task are aggregated before the task-agnostic optimization. Also, according to the sampling strategy of one task in the proposed method, one communication has time complexity $\Omega (1)$. In regard to this task sampling strategy, when we studied the impact of the number of sampling tasks to update the task-agnostic body, we found that one task sampling provides sufficient performance. We reported this in the response of the next comment C3.

---

> ### Author Response · Authors · 2021-11-21
> **Response to reviewer kCVH (1/4)**
>
> $#$**C1. There is no mention of any sort of privacy guarantees given the proposed architecture, despite "privacy-preserving" being part of the title. Privacy preservation is a strong claim that needs to be backed up rigorously. Moreover, in their ethics statement, the authors correctly layout that transmitted hidden features "may leak the raw data to some degree."**
>
> Thank you for the valuable comment. Although the data privacy attack can take place on the features transported between the server and clients, we use the term “privacy-preserving” because our model is designed to use distributed local data without sharing the data with other clients or a central device. In fact, this issue is one of the most difficult and common problems in distributed learning, including FedAvg, which is being actively studied in this area. To our best knowledge, split architecture is still considered safer than sharing the whole network weights as in federated learning for inversion attacks.
>
> Yet another powerful and unique mechanism for maintaining privacy in our method arises when the proposed method's client-side network has a skip connection between the head and the tail. In this case, the transported feature characteristic can contain very lossy information from the original data. This indicates that attackers may not be able to reconstruct original data using the transmitted hidden features of the proposed method. We discussed this data privacy guarantee in Section 3.3 of the main paper as well as in Appendix D.1.

---

> > ### Comment · Reviewer_kCVH · 2021-11-21
> > **Thanks for your various comments and the revision**
> >
> > I thank the authors for their valuable comments on my review and their revision. The revision has improved the submission substantially. My comments were addressed mainly due to the addition of Section 3.3 and the there referenced Appendix D. I believe that most of the other reviewers' comments were also addressed and can now recommend the submission for acceptance.
> >
> > Minor comment:
> > - The "Federated Learning" paragraph on pg. 3 contains an unresolved reference due to a typo.

---

> > > ### Author Response · Authors · 2021-11-21
> > > **Thanks for increasing your rating**
> > >
> > > Thank you for your positive feedback again and for increasing your rating.
> > > We have also updated the paper that reflects your minor comment.

---

### Official Review · Reviewer_fi2d · 2021-11-05

**Correctness:** 4
**Technical Novelty And Significance:** 3
**Empirical Novelty And Significance:** 2
**Recommendation:** 5
**Confidence:** 2

**Main Review:**

Strengths:
(i) the key idea of the paper is communicated effectively. The method is an extension of "SplitFed", Thapa et al. 2020 to multi-task learning with a shared backbone on the server side.
(ii) the experiments validate that the proposed method is viable, and achieves comparable performance to/marginal improvements over existing privacy-preserving training methods on the tasks considered in this work.
(iii) some synergy can be seen in the performance on various tasks due to joint multi-task training, whereby the proposed method outperforms end-to-end and single-task distributed training (table 2).

Weaknesses:
(i) the key weakness is that the "core contribution" of using a transformer (Vit) based task-agnostic backbone on the server has incremental novelty over the more standard/prior works (specifically, Thapa et al.). The key novelty is joint-training of a shared backbone on *different* tasks in the proposed work --- this is a natural extension of the prior method.

(ii) it is unclear why the specific decision of using CNNs as heads/tails (on the clients) and a ViT as backbone (on the server) is so crucial; could the roles be swapped --- ViT based heads/tails, and a CNN backbone? Overall this particular design choice has not been shown to be critical for operation or the performance of privacy-preserving distributed learning. Any differentiable neural network can be placed at the server/client sides as long as the two can be "plumbed" together.

(iii)  the number of clients in the "distributed" experiments are small (perhaps this is prevalent in this research community). For example, only 5 clients, are used for 4 tasks (section 4). Hence, it is difficult to gauge the practical deployment of this method, as common distributed learning issues like clients dropping out, asynchronous communication is not tested.





**Summary Of The Paper:**

This work is aimed at distributed "privacy preserving" training of neural networks for image processing tasks like deblocking, denoising, deraining, and deblurring. "One of the most important contribution" [pg 2] is breaking down the neural network model into task-specific convolutional head and tails (trained on "clients"), and a common shared (across tasks) Transformer based feature backbone, which is trained on the server. The heads/tails and the transformer backbone are trained in an alternate manner by assuming the other model to be fixed.

The proposed is similar to the method "Splitfed" (Thapa et al., 2020) but is extended for different tasks (as described above).

Experimental results demonstrate:
(i) successful training of the neural network models with the proposed method.
(ii) better/comparable performance to prior works on distributed/privacy-preserving methods.
(iii) better performance using the Vit backbone as compared to CNN backbones, and also with the proposed multi-task vs. single-task setting.

**Summary Of The Review:**

This work proposes an extension of the method "SplitFed" of Thapa et al. 2020, where a common transformer based backbone is trained across *different* tasks; this is a natural extension of the prior work. This method has been shown to achieve similar/marginally better performance across all the five tasks considered in this work. However, the scale of experimentation is small (only 5 clients), and idea itself carries incremental novelty. In view of the above, I can recommend this paper further, but perhaps with some reservation.

-------------------------------

Post authors' response:

Thank the authors for taking the time to address the concerns raised in the review. However, as detailed in individual comments, their response is not convincing. The key points are: (1) no fundamental reason to favor Transformers over other neural modules, (2) no guarantees of privacy preservation, despite claims to this effect, (3) no large-scale distributed study to validate their design. Hence, I urge the authors to address these, and revise the paper. I am unable to recommend this work further in the current form.

---

> ### Author Response · Authors · 2021-11-21
> **Response to reviewer fi2d (3/3)**
>
> $#$**C3. The number of clients in the "distributed" experiments are small (perhaps this is prevalent in this research community). For example, only 5 clients are used for 4 tasks (section 4). Hence, it is difficult to gauge the practical deployment of this method, as common distributed learning issues like clients dropping out, asynchronous communication is not tested.**
>
> We agree that 5 clients is a relatively small number. However, an important aspect of the proposed network architecture is that after training the Transformer body with a sufficient number of tasks and clients (in our case 5 clients and 4 tasks), additional training of the Transformer body is no longer required when a new client is added with a new task. Instead, all that is required is the training of customer-specific head/tails (see our new experiments in Appendix D.5). This makes the proposed method very practical, which can deal with asynchronous communication and dropout of the clients.
>
> In addition, even for the training phase of the task-agnostic Transformer body, we confirm that one client sample per iteration is sufficient to train the task-agnostic body so that the number of clients does not increase the communication costs for the task-agnostic learning. More precisely, only 10MB of data are transported per communication if the size and data type of the feature matrices correspond to our setting.
>
> We discussed this in Section 3.3 of the main paper, Appendix D.3, and D.4.

---

> > ### Comment · Reviewer_fi2d · 2021-11-27
> > **Comment**
> >
> > Thank you for sharing your response. If no-training, is required --- then one might just pre-train the backbone on the server offline (before deployment) --- so this is not an interesting use case.  Further, the effect of batch-size and stability of training based on that, and issues mentioned before which are tied to large-scale distributed deployment are not tested convincingly. Hence, this work is largely an academic exercise/tested in toy settings with no empirical study on the claims it makes, i.e., distributed privacy-preserving deployment.  Further, as also mentioned by other reviewers, there are no guarantees given for privacy-preservation --- it is not proved that one cannot recover information about the data from the gradients that are communicated. Hence, the claims in this work are not fully justified.

---

> > > ### Author Response · Authors · 2021-11-28
> > > **Respond to the comment**
> > >
> > > Thanks for responding to our reply. What we're saying is that one can get synergistic performance gains without training already pre-trained body that can be time-consuming. Also, in the point of privacy-preservation, we discussed the privacy-preserving property of the proposed method in Section 3.3 and Appendix D.1. In fact, another powerful and unique mechanism for maintaining privacy in our method arises when the proposed method's client-side network has a skip connection between the head and the tail. In this case, the transported feature characteristic can contain very lossy information from the original data. As shown in Appendix D.1, one cannot reconstruct data only using the transmitted hidden features of the proposed method.
> > >
> > > Furthermore, one may not recover the raw data even using the transported gradients. Although there is a study on inverting raw data from parameter gradients [1], this is related with the gradient of loss with respect to the network parameters that are transmitted in typical federated learning methods. On the other hand, our transported element is the gradient of loss with respect to the feature map, which does not allow the gradient inversion by [1]. Also, when the proposed method's client-side network has a skip connection between the head and the tail, the gradients are partially transported so that one cannot obtain the full gradients in the communication. Specifically, in the case of skip-connection, the gradient of loss $l_{c}$ with respect to the head output $f_{H}$ can be expressed as:
> > >
> > > $\frac{\partial l_c}{\partial \hat{y}}+\frac{\partial l_c}{\partial f_B}\frac{\partial f_B}{\partial f_H}$
> > >
> > > where $f_H$ and $f_B$ are the output feature map of the head and the body, respectively, $l_c$ is the task-specific loss on the client $c$, and $\hat{y}$ is the output of the tail. However, the first term $\frac{\partial l_c}{\partial \hat{y}}$ is not transported in our framework. Therefore, one may not recover the data through the transported gradients, as well as transported features.
> > >
> > > Nevertheless, since this work is more like a proof-of-the-concept study, rather than a ready-to-use solution for the industry, the issues of large-scale distributed deployment such as the computing power imbalance and privacy-related issues such as threatening privacy via inversion attack should be further analyzed for practical application. Still we have successfully implemented our framework in Flower which is a framework for the federated learning, and have verified that this is a working solution, although optimization is required for large scale deployment. We will add this in the final version (We cannot revise the paper in this period).
> > >
> > > [1] Geiping, Jonas, et al. "Inverting Gradients--How easy is it to break privacy in federated learning?." arXiv preprint arXiv:2003.14053 (2020).

---

> > > > ### Comment · Reviewer_fi2d · 2021-12-02
> > > > **comment**
> > > >
> > > > The authors have responded to 'recovering raw data' -- however, the comment was about 'information about data'. Raw data might not be recoverable, but some information (which might be enough to break anonymity) might leak through the transported gradients. No analysis / guarantees regarding this has been provided.

---

> > > > > ### Author Response · Authors · 2021-12-02
> > > > > **Respond to the comment**
> > > > >
> > > > > Thank you for the additional comment. Most of the existing federated / split learning frameworks focus on the confidentiality of the raw data. This is especially true for image processing, where the key point of privacy risk is the raw data itself rather than the extracted features or statistics. This is why we responded that raw data cannot be recovered. While we agree that the proposed framework does not guarantee any leakage of features or extracted statistics, we believe that this is also the case with most existing federated or split learnings. That being said, care should be taken with the anonymity of the identifiable information so that the transferred features or gradients do not reveal such information. This issue should be investigated in future research in federated learning.

---

> ### Author Response · Authors · 2021-11-21
> **Response to reviewer fi2d (2/3)**
>
> $#$**C2. It is unclear why the specific decision of using CNNs as heads/tails (on the clients) and a ViT as backbone (on the server) is so crucial; could the roles be swapped --- ViT based heads/tails, and a CNN backbone? Overall this particular design choice has not been shown to be critical for operation or the performance of privacy-preserving distributed learning. Any differentiable neural network can be placed at the server/client sides as long as the two can be "plumbed" together.**
>
> Thank you for your constructive comments. The reason for developing our model with CNN-based heads / tails and the Transformer-based body is to take advantage of each network. In particular, Transformer learns the global attention of the input sequence through self-attention modules and has recently been extensively studied for various computer vision tasks. One of the most unique advantages of Transformer is to convert "unattended" input feature vectors into "attended" output feature vectors by learning global attention and non-local interactions between the input features. Accordingly, the task-specific head / tail can be only trained to learn task-specific local features, whereas the global feature can be learned through the Transformer body. This disentangled representation of local and non-local features has been pursued throughout the development of deep networks [1,2,3]. Thus, we argue that the proposed Transformer-based approach is considered to be one of the most advanced architectures for achieving this goal, as it improves synergistically overall performance, as shown in our experiments, and at the same time leads the privacy-preserving split-learning architecture.
>
> To verify the synergistic effects of the Transformer body that is suitable for distributed learning, we performed an additional experiment by replacing the Transformer body with a CNN model.  In particular, we configured the CNN body with C-B-R blocks, where C is a convolution layer, B is a batch normalization layer, and R is a ReLu layer. For a fair comparison, we've set the number of C-B-R blocks so that they have almost the same number of learnable parameters as the Transformer body we proposed. Then, using this CNN body, we implemented our proposed task-specific and task-agnostic learning for the multiple image processing tasks similar to our method in the main paper. As a result, Table 1 (below) shows that our method with the Transformer body achieves higher performance in distributed learning with multiple tasks compared to the model with the CNN body. We added this study to Appendix D.2.
>
> Furthermore, we have also applied our model to additional natural image inpainting and medical image denoising tasks. By fixing the Transformer body, which was pre-trained for the four tasks in the main paper, we only trained the task-specific heads and tails on clients. Even without fine-tuning the Transformer body, our task-specific networks offer better image quality compared to the model that consists of just a CNN-based encoder and decoder. This implies that the Transformer body plays a significant role in learning a common feature representation for different tasks and also provides sufficient capacity for image processing of different image domains such as medical images. See Appendix D.5.
>
> [1] Ye, J.C., Han, Y. and Cha, E., 2018. Deep convolutional framelets: A general deep learning framework for inverse problems. SIAM Journal on Imaging Sciences, 11(2), pp.991-1048.
>
> [2] Zhang, S., He, X. and Yan, S., 2019, May. Latentgnn: Learning efficient non-local relations for visual recognition. In International Conference on Machine Learning (pp. 7374-7383). PMLR.
>
> [3] Wang, X., Girshick, R., Gupta, A. and He, K., 2018. Non-local neural networks. In Proceedings of the IEEE conference on computer vision and pattern recognition (pp. 7794-7803).

---

> > ### Author Response · Authors · 2021-11-21
> > **Response to reviewer fi2d (2/3): Table**
> >
> > **Table 1.** Results of study on the effect of the Transformer body of TAViT. The average values of PSNR / SSIM are displayed.
> >
> > |       Task      |       Input       |  Cycle 0.5 (CNN body) |  Cycle 1.0 (CNN body) | Cycle 0.5 (Transformer body) | Cycle 1.0 (Transformer body) |
> > |:---------------:|:-----------------:|:----------------:|:----------------:|:-----------------------:|:-----------------------:|
> > | Deblocking (Q10) |  25.67 dB / 0.719 | 26.42 dB / 0.752 | 26.49 dB / 0.755 |     27.53 dB / 0.781    |    **27.57 dB / 0.782**    |
> > | Deblocking (Q50) | 31.51 dB / 0.902 | 32.92 dB / 0.921 | 32.95 dB / 0.921 |     32.92 dB / 0.921    |    **33.01 dB / 0.922**    |
> > |     Denoising    |  19.03 dB / 0.336 | 30.22 dB / 0.866 | 30.50 dB / 0.866 |    30.57 dB / 0.868    |    **30.62 dB / 0.869**    |
> > |  Deraining (RainH) | 13.55 dB / 0.380 | 28.23 dB / 0.851 | 28.26 dB / 0.852 |    28.24 dB / 0.860    |    **28.75 dB / 0.862**    |
> > |  Deraining (RainL) | 26.90 dB / 0.839 | 31.58 dB / 0.930 | 31.77 dB / 0.930 |    **33.17 dB / 0.939**    |    32.69 dB / 0.936    |
> > |      Deblurring     |  25.64 dB / 0.790 | 28.78 dB / 0.865 | 28.94 dB / 0.868 |    28.94 dB / 0.871    |    **29.09 dB / 0.873**    |

---

> > > ### Comment · Reviewer_fi2d · 2021-11-27
> > > **Comment**
> > >
> > > Thank the authors for explaining their rationale behind favoring a Transformer as the backbone, and augmenting their experiments with additional ones offering comparisons with ConvNet based backbones. While Transformers do outperform ConvNets marginally, they are not crucial for the design or working of the proposed method. Hence, the original comment of being able to use any differential neural module stands, and the necessity of Transformers only (which is strongly implied in the text and also the title of this work) is not justified convincingly. I would urge the authors to rethink the core contributions of their work.

---

> > > > ### Author Response · Authors · 2021-11-28
> > > > **Respond to the comment**
> > > >
> > > > Thanks for the constructive comment. We design a multi-task distributed learning framework using Transformer and show the validity of the design, not just propose a Transformer-only framework.  As we responded in the above rebuttal, the reason we design our model with Transformer body is to convert task-specific feature maps into task-agnostic feature maps through the property of Transformer that learns global attention between the input features. We described this in Section 1 of the revised paper.

---

> ### Author Response · Authors · 2021-11-21
> **Response to reviewer fi2d (1/3)**
>
> $#$**C1. The key weakness is that the "core contribution" of using a transformer (Vit) based task-agnostic backbone on the server has incremental ‘novelty’ over the more standard/prior works (specifically, Thapa et al.). The key novelty is joint-training of a shared backbone on different tasks in the proposed work --- this is a natural extension of the prior method.**
>
> At first glance, our method may appear similar to SplitFed (Thapa et al) as we split task-specific networks. However, there exists fundamental differences that make our method sufficiently novel.
>
> First, we focus on multi-task distributed learning while the existing distributed learning methods typically focus on single tasks with different datasets. In particular, SplitFed (Thapa et al) has only a head on each client and a shared tail located on the server so that the SplitFed server requires the client labels to compute gradients and update the networks. This can be a serious problem in distributed learning in terms of data privacy, since in many practical situations (such as medical data), the labels themselves are precious resources that data owners do not want to share. Furthermore, SplitFed generates outputs from the tail, so in order to apply to multi-task learning, the server should have all associated labels, which makes the SplitFed more prone to data leakage. On the other hand, our model enables distributed multi-task learning without sending raw data and their labels to the other devices, which makes our method more secure.
>
> Second, the proposed split architecture, where the server has a shared Transformer-based  backbone and each client has its own head and tail networks, is unique and has many advantages. Our approach is not only safer, as previously discussed, but the model can also learn the common representations of different tasks, which, thanks to the global attention mechanism in the Transformer body, leads to an increase in synergy in multi-task learning. Once the common representation is learned, the Transformer body is universal in the sense that we can add new head and tail networks for different tasks without fine-tuning the Transformer body, as shown in the newly added experiments in Appendix D.5.
>
> We emphasized this novelty over the prior split learning methods in Abstract and Introduction.

---

### Author Response · Authors · 2021-11-21
**Message to reviewers**

Dear Reviewers,

We sincerely appreciate your time and efforts in reviewing our paper, and your constructive and insightful comments. We have faithfully responded to your comments and did our best to provide additional experimental results per your suggestions. The main paper is also updated, and the changes are mainly in Sections 3.3, Appendix D. We would be glad to discuss any further questions you may have after reviewing our responses.

Best regards,

The authors

---

### Decision · Program_Chairs · 2022-01-20

**Decision:**

Reject

**Comment:**

The paper aims to devise a distributed multi-task privacy preserving framework for image processing. In this regard, author propose partitioning neural network models into task specific heads/tails and a common task-agnostic feature backbone (body). A training procedure is designed which is claimed to be privacy preserving wherein the head and tail is trained locally on the client or using federated learning when multiple clients share a task, while the main backbone/body is trained in a centralized manner by collecting appropriate gradients from the clients. Making easy to follow code is also highly appreciated. We thank the reviewers and authors for engaging in an active discussion and also updating the paper. While the new version is definitely resolves some of the concerns of the reviewers, some still remain. Privacy preserving in title and in main body of the paper seems misleading. Proposed method doesn't provide any guarantees for privacy (also pointed out by many reviewers). The author response doesn't seem to be convincing and other federated learning papers do not claim privacy unless having some specific mechanism like adding noise, secure aggregate, etc. Also, the reviewers are in consensus that novelty as well as large scale empirical evaluation is limited.